# The multifaceted role of the viral 2A protease in enterovirus replication and antagonism of host antiviral responses

Jelle G. Schipper[1☯], Chiara Aloise[1☯], Sereina O. Sutter[1], Marleen Zwaagstra[1], Arno L.W. van Vliet[1], Rana Abdelnabi[2,3], Bob Ignacio[4], Kimberly M. Bonger[4¤a], Dagmar Roelofs[5], Judith M.A. van den Brand[5], Richard W. Wubbolts[6], Lucas J.M. Bruurs[7¤b], Hendrik Jan Thibaut[2], Johan Neyts[2,3], Marvin E. Tanenbaum[7,8‡*], Frank J. M. van Kuppeveld[1‡*]

1 Section of Virology, Division of Infectious Diseases and Immunology, Department of Biomolecular Health Sciences, Faculty of Veterinary Medicine, Utrecht University, Utrecht, the Netherlands, 2 Department of Microbiology, Immunology and Transplantation, Rega Institute, Virology, Antiviral Drug & Vaccine Research Group, KU Leuven, Leuven, Belgium, 3 KU Leuven - Department of Microbiology, Immunology and Transplantation, VirusBank Platform, Leuven, Belgium, 4 Department of Synthetic Organic Chemistry, Chemical Biology Lab, Radboud University, Nijmegen, the Netherlands, 5 Division of Pathology, Department of Biomolecular Health Sciences, Faculty of Veterinary Medicine, Utrecht University, Utrecht, the Netherlands, 6 Center for Cell Imaging, Department of Biomolecular Health Sciences, Faculty of Veterinary Medicine, Utrecht University, Utrecht, the Netherlands, 7 Oncode Institute, Hubrecht Institute–KNAW and University Medical Center Utrecht, Utrecht, the Netherlands, 8 Department of Bionanoscience, Delft University of Technology, Delft, the Netherlands

☯ These authors contributed equally to this work.
‡ These authors share senior authorship.
¤a Current address: Department of Chemical Biology & Immunology, Leiden Institute of Chemistry, Leiden University, Leiden, the Netherlands
¤b Current address: Bilthoven Biologicals B.V., Antonie van Leeuwenhoeklaan 9, AL Bilthoven, The Netherlands
* f.j.m.vankuppeveld@uu.nl (FJMvK); m.tanenbaum@hubrecht.eu (MET)

## Abstract

Enteroviruses dramatically remodel the cellular infrastructure for efficient replication and curtailing host antiviral responses. The roles of viral proteins in these processes have been studied mostly *in vitro*, by ectopic overexpression, or by surrogate infection systems, all of which have shortcomings. Here, we replace the essential 2A cleavage site at the P1-P2 junction with an internal ribosome entry site (IRES), 3CD cleavage site, or T2A sequence, allowing us to catalytically inactivate 2A$^{pro}$ in the virus context. Viruses with an inactive 2A$^{pro}$ are hampered in replication in cell lines and are severely attenuated in a Coxsackievirus B3 (CVB3) mouse pancreatitis infection model. We show that 2A$^{pro}$ is essential for disturbing nucleocytoplasmic transport, shutting down host mRNA translation, suppressing stress granule formation, suppressing the induction of the IFN response, and overcoming IFN-induced restriction factors. Moreover, using an advanced single-molecule live cell imaging approach, we reveal that 2A$^{pro}$ is important for the initial round of replication of the incoming viral RNA, which is a bottleneck for efficient infection. Thus, 2A$^{pro}$ plays a critical role in

**Data availability statement:** Raw data is available via Mendeley Data at doi: 10.17632/zrgfy9wvr4.1. Due to size limitations, for the time lapse imaging data representative raw pictures are made available for download. The entire dataset can be accessed upon request from the secretariat of the Virology section of Utrecht University at Virol902@uu.nl.

**Funding:** This work was financially supported by an NWO Klein-2 Grant (OCENW.KLEIN.344) to M.E.T. and F.J.M.v.K; a European Commission (EC) Marie Sklodowska-Curie Actions (MSCA) Innovative Training Network H2020-MCSA-ITN-2019 (INITIATE, Grant Nr. 813343) to F.J.M.v.K; an EC ERC Advanced Grant to F.J.M.v.K (virLUMINOus, Grant Nr. 101053576); an EC ERC Consolidator Grant (VirIm, Grant Nr. 101044794) to M.E.T; the Belgian Federal Government for the VirusBank Platform to J.N.; and a EC MSCA Postdoctoral Fellowship (BARONET, Grant Nr. 101152667) to S.O.S. Furthermore, M.E.T. was financially supported by the Oncode Institute, which is funded in part by the Dutch Cancer Society (KWF). The funders had no role in study design, data collection and analysis, decision to publish, or preparation of the manuscript.

**Competing interests:** The authors have declared that no competing interests exist.

subverting antiviral responses and establishing a favorable environment to expedite enterovirus replication.

---

## Author summary

Enteroviruses are a group of viruses within the family *Picornaviridae* that pose a serious threat to both human and animal health. The prototypical example is poliovirus, but the genus also includes coxsackieviruses, rhinoviruses, and emerging numbered enteroviruses such as EV-A71 and EV-D68. During infection enteroviruses dramatically remodel their host environment to facilitate efficient replication, and to counteract cellular antiviral defense mechanisms. These modifications are thought to be largely mediated via host cell cleavage events by the two viral proteases, 2A$^{pro}$ and 3Cpro. The essential role both proteases also play in viral polyprotein processing however precludes their catalytic inactivation to study their relative importance and contributions. Consequently, most of our knowledge concerning the two proteases comes from in vitro studies, ectopic overexpression, and surrogate infection systems. Here, we replaced the essential 2A$^{pro}$ cleavage site at the P1-P2 junction with alternative genetic elements to mediate separation of the polyprotein independently from 2Apro, allowing us to catalytically inactivate 2A$^{pro}$ within the virus context. We show the importance of 2A$^{pro}$ for efficient replication both in cell culture and in vivo. Furthermore, with a wide array of approaches, we demonstrate that 2Apro, and not 3Cpro, plays a critical role in host adaptation and counteracting the host response. Finally, we make use of an advanced single-molecule live cell imaging approach to dissect the role of 2A$^{pro}$ in viral replication.

## Introduction

Enteroviruses (EVs) are non-enveloped viruses with a single-stranded (ss) RNA genome of positive polarity [(+)ssRNA] that belong to the family *Picornaviridae*. The genus *Enterovirus* includes major human pathogens, such as poliovirus (PV), coxsackievirus (CV), echovirus, emerging numbered enteroviruses like EV-A71 and EV-A68, and rhinoviruses (RVs). Overall, there are over 150 serotypes of non-polio enteroviruses and more than 170 rhinovirus serotypes [1], hence vaccination is not a feasible option for all of them. Consequently, there is an urgent need for broad-spectrum antiviral drugs against enteroviruses, the development of which requires a comprehensive understanding of the enterovirus life cycle. While different enteroviruses use distinct receptors and entry pathways, many post-entry steps are conserved. After accessing the cytosol, the genomic viral RNA (vRNA) is directly translated into a single large polyprotein. As opposed to canonical cap-dependent translation initiation, translation of the viral ORF is initiated via an internal ribosome entry site (IRES). The resulting polyprotein is auto-proteolytically processed

by viral proteases (2A[pro] and 3C[pro]) yielding 11 distinct proteins, including the viral RNA-dependent RNA polymerase, viral capsid proteins, and several other viral proteins involved in replication and/or triggering alterations in the host cell environment. After the initial synthesis of viral proteins, vRNA replication starts with the formation of a negative-strand copy of the incoming viral genome to generate a double-stranded (ds)RNA replication intermediate. Notably, cytosolic dsRNA is highly immunogenic, as it can be recognized by cellular sensors that activate antiviral response pathways [2]. The negative-strand copy next serves as a template for synthesis of multiple new (+)ssRNA strands, which either start subsequent rounds of translation and replication or, at late stage of infection, are encapsidated to form new infectious virus particles [3].

To optimize the cellular environment for viral translation and replication, and to suppress cellular antiviral defense mechanisms, viruses have evolved elaborate strategies to perturb cellular signaling and functions. For example, enteroviruses rapidly inhibit cap-dependent host mRNA translation [4,5]. Traditionally, this activity was considered to be important for redirecting resources for optimal translation of viral RNA (vRNA), but this activity may also be important to suppress *de novo* synthesis of proteins with antiviral function [4,6]. Moreover, enteroviruses disturb the integrity of the nuclear pore complex (NPC), thereby triggering a so-called nucleocytoplasmic trafficking disorder (NCTD). This activity results in the translocation of cellular RNA binding proteins to the cytosol, such as La-autoantigen and Pyrimidine-tract binding protein (PTB) [7–9], which are thought to enhance key viral processes such as vRNA translation and genome replication. Moreover, it results in the nuclear retention of host mRNA, proposedly to suppress cellular antiviral signaling [6,10]. Furthermore, enteroviruses suppress the formation of stress granules (SGs), which are considered to serve as a platform for the assembly and integration of several antiviral signaling pathways, including the integrated stress response (ISR) and type I interferon response (IFN-α/β) pathways. Apart from this, enterovirus infection results in the cleavage of dsRNA sensors and downstream signaling molecules involved in the induction of IFN-α/β production.

Both 2A[pro] and 3C[pro] have been attributed central roles in the viral strategies to optimize vRNA translation and replication, and antagonize the host antiviral responses described above. However, as of yet, we know little about the relative importance and individual contribution of each protease to these strategies. Most studies were conducted using either *in vitro* experiments, ectopic overexpression, or surrogate infection models, such as encephalomyocarditis virus (EMCV) carrying inactivating mutations in the Leader (L) protein (EMCV-L(Zn)) expressing 2A[pro] as a heterologous protein. Unfortunately, the ectopic expression of the protease isolated from all other viral proteins does not accurately reflect the context and spatio-temporal dynamics of infection. Although the surrogate system does entail an infection scenario, it is of another virus and leaves open the question of whether the observed effects occur in EV-infected cells and if they are relevant. Thus, the study of viral proteins in isolation or out of the infection setting of their parent virus can give erroneous or difficult-to-interpret insights into their true function. This is exemplified by studies towards the function of the L protein of Crimean Congo Hemorrhagic Fever virus (CCHFV), whose predicted deubiquitinase (DUB) domain was shown to inhibit IFN signaling when expressed in isolation, but upon inactivation in the virus was identified to be involved in cleaving off ubiquitin *in cis* from L to regulate its function, rather than to suppress IFN signaling [11].

The critical role that both proteases play in polyprotein processing has largely precluded the possibility of catalytically inactivating them, and thereby to study their individual contribution to the virus-host interaction in the relevant infection context. Previously, mutants of PV and EV-A71 were reported where 2A[pro] had been either removed entirely or catalytically inactivated, enabled in both cases by the introduction of a second IRES in between P1-P2 [12,13]. This resulted in severe attenuation, but other than assessing the role of 2A[pro] for SG formation [12], these viruses were not characterized in great detail. Here, we report the construction of viable CVB3, in which we replaced the 2A[pro] cleavage site at the VP1/2A junction with either an additional IRES, a T2A self-cleaving peptide, or a 3 CD cleavage site (3 CD$_{cs}$), enabling mutational inactivation of 2A[pro]. CVB3 containing a catalytically-inactive 2A[pro] (CVB3–2Amut) exhibits only a relatively minor reduction in fitness in some cell types, but is nearly completely restricted in others. Furthermore, CVB3–2Amut is severely attenuated *in vivo* in a CVB3 pancreatitis mouse model, illustrating the important role that 2A[pro] plays during

infection. Detailed characterization of the consequences of 2A[pro] inactivation in CVB3 infected cells reveals that 2A[pro], but not 3C[pro], plays key roles in suppressing host mRNA translation, NCTD induction, suppressing SGs, and inhibiting IFN-α/β responses. Employing VIRIM (Virus Infection Real-time Imaging), an advanced single-molecule live cell imaging assay that we recently developed [14], we show that 2A[pro] inactivation affects the dynamics of the early phases of the enterovirus life cycle and interferes with the ability to progress through an early critical phase, namely replication of the incoming viral genome. In conclusion, we identify the distinctive role of 2A[pro] in promoting viral translation, replication, modification of host cell functions, and suppression of antiviral responses in enterovirus-infected cells.

## Results

### Design, generation, and growth kinetics of recombinant CVB3 with inactive 2A[pro]

To engineer recombinant CVB3 viruses with catalytically inactive 2A[pro], we replaced the 2A[pro] cleavage site at the P1-P2 junction with either (i) the IRES sequence of Theiler's encephalomyelitis virus (TMEV), (ii) a self-cleaving peptide from *Thosea asigna* virus (T2A), or (iii) a sequence corresponding to the 3 CD cleavage site recognized by 3C[pro]/3 CD[pro] (3 CD$_{cs}$) (Fig 1A). All these elements effectively mediated separation of P1 and P2 (S1C Fig), alleviating the critical role of 2A[pro] in polyprotein processing and allowing for its catalytic inactivation. Specifically, mutations were introduced in the catalytic triad (H21A, D39A, and C110A) to abolish 2A[pro] activity (these will further be referred to as 2Amut). Recombinant CVB3 viruses were recovered upon transfection of human embryonic kidney 293-T (HEK293-T) cells with *in vitro* RNA transcripts obtained from cDNA clones.

To assess the growth kinetics of the CVB3 with inactive 2A[pro], single-cycle growth curves were performed on HeLa cells, measuring both infectious particles (TCID$_{50}$/ml) and vRNA levels. A full infection cycle of wt-CVB3 on HeLa cells takes roughly 8–10 hours. All three 2A-wild type (wt) viruses replicated well, albeit with a slight reduction in resulting vRNA levels and infectious virus titers compared to wt-CVB3, especially for CVB3-T2A-2Awt. This suggests that the introduction of alternative elements at the P1-P2 junction generally effects separation, yet with somewhat differing effects on viral replication efficiency (Fig 1B). Notably, all CVB3–2Amut viruses showed a delay in vRNA replication and a reduction in virus production compared to 2Awt viruses (Fig 1C).

To investigate in more detail the impact of 2A[pro] inactivation on vRNA translation and replication, we transfected HeLa and HEK293-T cells with equivalent amounts of *in vitro* transcribed RNAs from CVB3–3 CD$_{cs}$-2Awt and 2Amut cDNAs containing the *Renilla* luciferase (RLuc) upstream of the capsid coding region. The increase in luciferase signal that can be observed after 5 hours (hrs), which reflects translation of newly synthesized vRNAs, was severely reduced by 2A[pro] inactivation (Fig 1D). In contrast, no large differences were observed in the luciferase levels produced in the first hrs after transfection, before substantial replication has taken place, implying that 2A[pro] has little effect on the efficiency of translation on the incoming vRNA. Together, these data highlight the essential role of the proteolytic activity of 2A[pro] for efficient vRNA replication and subsequent virus production.

The somewhat modest reduction in viral growth observed in HeLa cells prompted us to explore whether this might be more pronounced in different cell lines. To this end, we first titrated our CVB3–3 CD$_{cs}$ and T2A virus stocks on various cell lines and assayed the development of cell death. Comparison of CVB3–3 CD$_{cs}$-2Awt and -2Amut revealed that, whereas 2A[pro]-inactivated virus replicated relatively well in HeLa cells, replication in Buffalo green monkey (BGM) kidney cells, African green monkey kidney Vero-E6 cells, and human lung carcinoma A549 cells was severely impaired (S1B Fig). Consistently, infectious virus production was shown to be severely restricted on BGM, Vero, and A549 when assessed by titration on susceptible HeLa cells (Figs 1E and S1C). The reason for the divergent importance of 2A[pro] catalytic activity in different cell lines remains unclear, but it might relate to steady-state expression levels of specific intrinsic restriction factors that are countered by 2A[pro]. Alternatively, 2A[pro] activity might be more or less important to facilitate a pro-viral host environment in different cell lines. As we observed only minor differences between the CVB3-IRES, -3 CD$_{cs}$, and -T2A viruses, we used

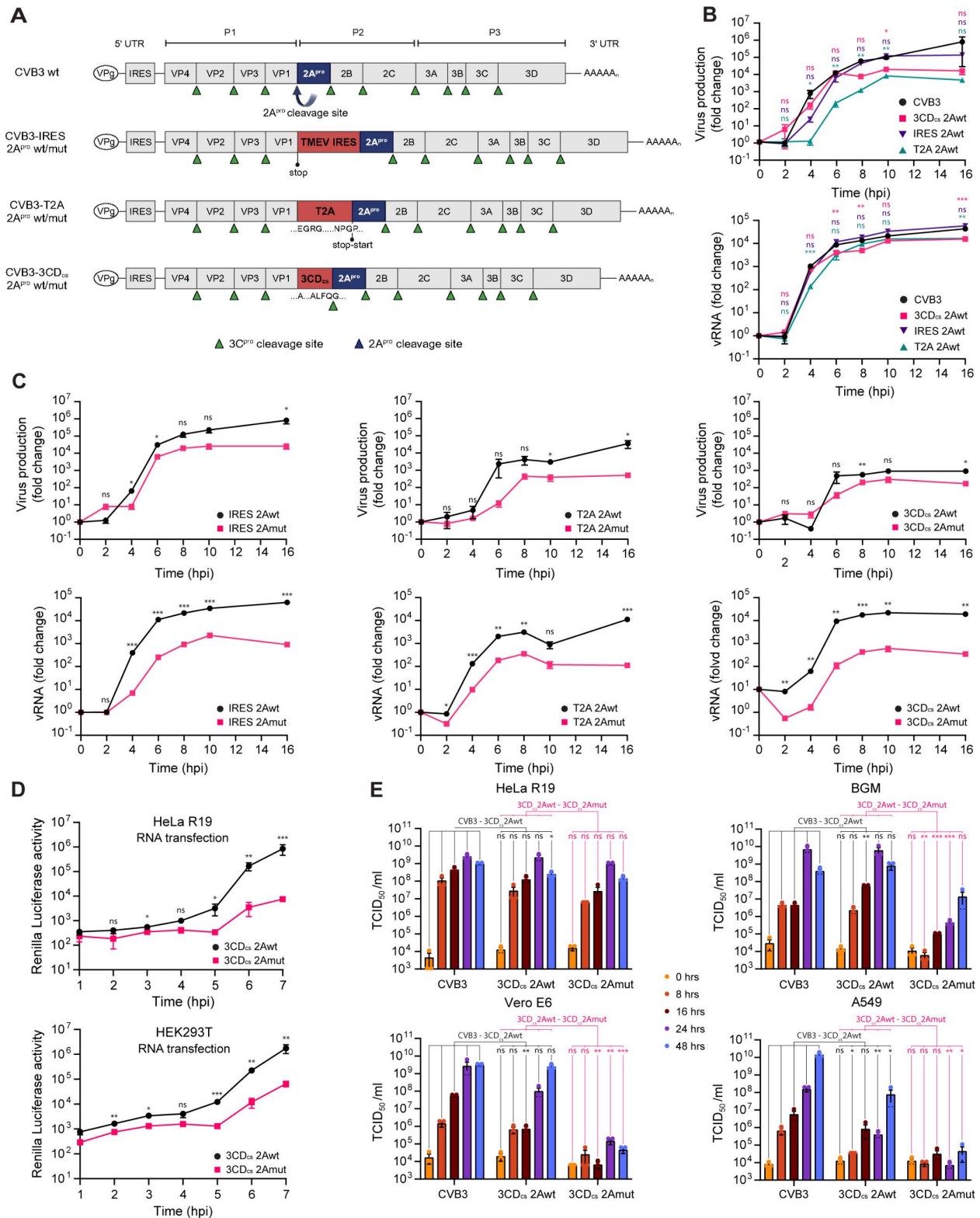

**Fig 1. Design and characterization of Coxsackievirus B3 with catalytically-inactive 2A protease (2A^pro).** (A) Schematic representation of the wild-type (wt) Coxsackievirus B3 (CVB3) genome and its engineered derivative viruses with inactivated 2A^pro (2Amut). The 2A^pro cleavage site was replaced

by a Theiler's murine encephalomyelitis virus (TMEV)-derived IRES sequence, a T2A peptide, or a sequence corresponding to the cleavage site of 3C/3 CD protease (3C$^{pro}$) (3 CD$_{cs}$). Specific cleavage sites of the viral proteases are indicated with blue (2A$^{pro}$) and green (3C$^{pro}$) triangles. (B) Growth kinetics of recombinant 2A$^{pro}$ wt viruses (3 CDcs-2Awt, T2A-2Awt and IRES-2Awt) in HeLa-R19 cells infected at a multiplicity of infection (MOI) of 5, as measured by TCID$_{50}$ assay (upper panel) or qPCR (lower panel). (C) Growth kinetics of the different 2Awt and 2Amut viruses in HeLa-R19 cells. Experiment was performed as in panel (B). (D) HeLa-R19 or HEK293-T cells were transfected with equal amounts of in vitro transcribed RNA of Renilla-luciferase (RLuc)-3 CDcs-2Awt or -2Amut, and luciferase levels were determined at the indicated times post infection. (E) Growth curves of 2Awt and 2Amut viruses obtained from infected HeLa-R19, BGM, Vero E6 and A549 cells. Data represent the mean ± SEM of one representative experiment (B-D) or three independent experiments (E). The illustration in (A) was created in BioRender. Schipper, J. (2025) https://BioRender.com/7bqvkpe. Statistical significance was assessed by multiple two-tailed unpaired t-tests with multiple comparisons corrections using the Bonferroni-Dunn method (* p < 0.05; ** p < 0.01; *** p < 0.001).

the CVB3–3 CD$_{cs-}$2Awt and corresponding 2Amut viruses in further experiments. These viruses will further be referred to in the text as CVB3–2Awt and CVB3–2Amut, respectively.

## 2A$^{pro}$ activity is indispensable for CVB3 virulence and replication in a murine model

The more severe attenuation we observed in certain cell lines with CVB3–2Amut prompted us to investigate the virulence of this mutant virus in an *in vivo* setting. To this end, we employed a well-established CVB-induced pancreatitis mouse model [15,16]. SJL mice (n = 4 per group) were infected intraperitoneally with equivalent amounts (either 1.0*10$^7$ TCID$_{50}$/ml or 5.0*10$^5$ TCID$_{50}$/ml) of CVB3, CVB3–2Awt, or CVB3–2Amut. At 3 days postinfection - i.e., the time point at which a peak of pancreatic lesions can be observed - the mice were sacrificed and the pancreas was removed for analysis (Fig 2A). Infectious virus titers were determined by endpoint titration and vRNA levels quantified using RT-qPCR. Furthermore, using a standardized scoring system, we assessed histopathological damage (i.e., necrosis, inflammation, and edema). Virus titers and vRNA levels in the pancreas of mice infected with CVB3–2Awt were lower (10- or 100-fold, respectively) and histopathological damage was less severe than in mice infected with CVB3 (Fig 2B and 2C). This may be explained by a delay in replication of CVB3–2Awt in some cell types, as observed in A549, BGM, and Vero cells (Fig 1B). CVB3–2Amut, however, was severely impaired in replication. Infection with this virus resulted in 1000-fold less vRNA in the pancreas compared to CVB3–2Awt, no infectious virus could be isolated from the pancreas, and no histopathological damage was observed. Thus, CVB3–2Amut is severely attenuated *in vivo*, emblematic of the pivotal role which 2A$^{pro}$ plays in many aspects of the viral life cycle. To unravel this further, we resorted to studying CVB3–2Amut in HeLa cells, as this cell line supported replication relatively well.

## 2A$^{pro}$ is essential for disrupting nucleocytoplasmic trafficking in CVB3-infected cells

Enteroviruses profoundly disrupt active nucleocytoplasmic transport, which is regulated by the NPC, resulting in an NCTD. Nuclear import mediated by several different types of nuclear localization signals (NLSs), as well as the classical nuclear export pathway are affected [6], leading to the cytosolic relocalization of several nuclear-resident host factors, e.g., RNA binding proteins, to enhance viral RNA translation and/or replication [6]. Furthermore, this results in the defective transport of (signaling) proteins into the nucleus and a block in mRNA export, thereby impairing antiviral response activation [10,17]. Both 2A$^{pro}$ and 3C$^{pro}$ have been implicated in triggering the NCTD by cleaving several phenylalanine-glycine (FG)-repeat-nucleoporins, which interact with each other to maintain the permeability barrier of the NPC. The initial induction of the NCTD correlates temporally with the cleavage of Nup98, which is mediated exclusively by 2A$^{pro}$ [17–21]. Later during infection, other Nups have been reported to be targeted as well by both 2A$^{pro}$ (Nup62, Nup153, and Nup358) [10,17,19,21–25], and by 3C$^{pro}$ (Nup153, Nup214, and Nup358) [20,26], but the extent to which these contribute to the NCTD is unknown.

To assess the relative importance of 2A$^{pro}$ for causing the NCTD, we performed Western blot (WB) analysis (Fig 3A). Importantly, we observed accumulation of viral proteins 2BC and 2C (Fig 3A) for all viruses, albeit with a slight delay in CVB3–2Amut infected cells. Interestingly, compared to CVB3 and CVB3–2Awt infected cells, 2A$^{pro}$ accumulation was

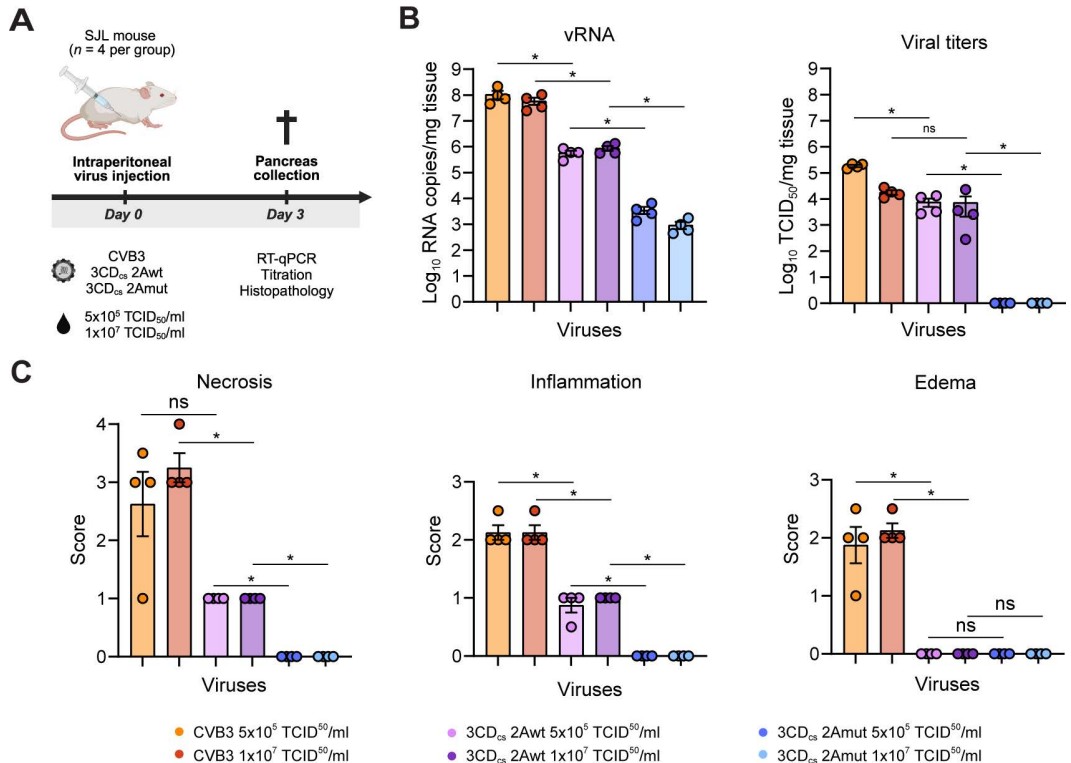

**Fig 2. 2A^pro activity is essential for CVB3 replication and pathogenesis in vivo.** (A) Schematic representation of the experimental procedure. SJL mice ($n = 4$ per group) were infected intraperitoneally with equivalent viral doses (either $1.0*10^7$ TCID$_{50}$/ml or $5.0*10^5$ TCID$_{50}$/ml) of CVB3, CVB3–2Awt or CVB3–2Amut. Mice were sacrificed three days post-infection for pancreas isolation and processing. (B) Viral RNA levels and infectious viral titers in the pancreas of infected mice were determined by RT-qPCR and end-point titration, respectively. (C) Pancreatic histopatology was assessed using a standardized scoring system ranging from 0 to 4 for necrosis (0 = not affected, 1 = < 10% of tissue affected, 2 = 11–25% of tissue affected, 3 = 26–50% of tissue affected, 4 = more than 50% of tissue affected), and from 0 to 3 for inflammation and edema (0 = none, 1 = mild, 2 = moderate, 3 = severe). Analyses were conducted without knowledge of the experimental groups. Statistical signifance was determined by Mann-Whitney $U$ test. Differences were considered significant if $p < 0.05$. The illustration in (A) was created in BioRender. Schipper, J. (2025) https://BioRender.com/0oodpvz.

markedly lower in CVB3–2Amut infected cells (S2A Fig). Cleavage products of Nup98 were detected in CVB3 and CVB3–2Awt infected cells from 2 hours post infection (hpi) onwards, with complete cleavage at 6 hpi. CVB3–2Amut was unable to cleave Nup98, even at later time points (Fig 3A), in agreement with earlier observations that 3C^pro does not cleave this Nup [20]. Similarly, using the general anti-FG-nup mAb-414 we demonstrate that 2A^pro cleaves a host of other FG-Nups, including Nup62, Nup153, and Nup214, but this occurs only at later time points [10,22,23]. Moreover, we failed to observe any cleavage of Nups at the hands of 3C^pro during CVB3–2Amut infection.

To corroborate the correlation between 2A^pro-mediated Nup cleavage and the induction of NCTD, we performed immunofluorescence (IF) analysis on HeLa cells stably expressing green fluorescent protein (GFP) fused to three tandem NLSs (HeLa-3xNLS-GFP) (Fig 3B). Cells infected with CVB3–2Awt showed rapid leakage of 3xNLS-GFP and heterogeneous nuclear ribonucleoprotein K (hnRNPK), a nuclear resident protein, from the nucleus to the cytosol. Moreover, we observed an altered nuclear morphology and less pronounced Nup98 signal in the nuclear rim over time, which was not observed in cells infected with CVB3–2Amut, emphasizing the essential role of 2A^pro in inducing NCTD and the dispensable contribution of 3C^pro. Of note, the failure to trigger NCTD in the CVB3–2Amut-infected cells is not due to a lack of 3C^pro activity, as evidenced by the observation of viral protein expression in CVB3–2Amut-infected cells from 4 hpi, albeit it at a lower level than in CVB3–2Awt-infected cells (Fig 3A), and from the observation of other 3C^pro-mediated cleavage events, such as

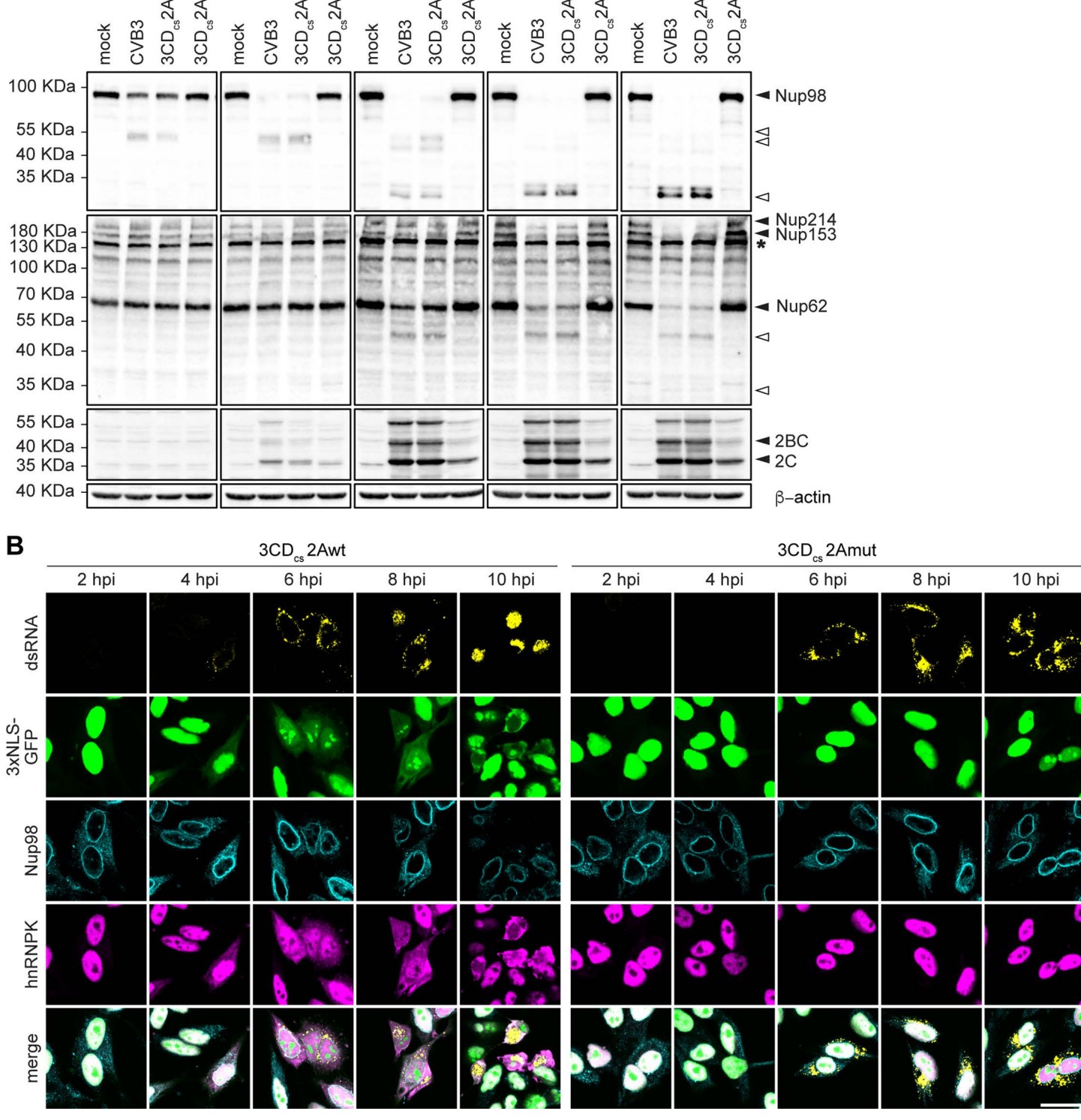

**Fig 3. 2A^pro plays a critical role in disrupting nucleocytoplasmic trafficking in CVB3 infected cells.** (A) Western blot analysis for Nup98, FG-repeat-containing nucleoporins, CVB3 2C protein and β-actin. HeLa-R19 cells, (mock-)infected with CVB3, 3 CD_cs-2Awt, or 3 CDc_cs-2Amut at an MOI of

10, were lysed at 2, 4, 6, 8 and 10 hpi. Black triangles denote intact proteins, white triangles indicate cleavage products, asterisk refers to a non-specific band which has been observed before with this antibody [18]. For densitometric analysis of all western blot data, see S1 Data. (B) Nucleocytoplasmic trafficking disorder (NCTD) is abolished in 3 CD$_{cs}$-2Amut infected cells. HeLa-3xNLS-GFP were infected at MOI of 10 with CVB3, 3 CD$_{cs}$-2Awt, and 3 CD$_{cs}$-2Amut. Cells were fixed with 4% paraformaldehyde at 2, 4, 6, 8, 10 hpi and analyzed by IF analysis for dsRNA as a marker of infection, for Nup98 as marker of the NPC, for hnRNPK and 3xNLS-GFP localization as indicators of NCTD. Scale bar: 25 µm.

that of Ras GTPase-activating protein-binding protein 1 (G3BP1) and RIG-I, in the same experimental lysates (see Figs 4D and 5B). Together, our data present evidence that 2A$^{pro}$ plays a critical role in triggering the NCTD in infected cells and argue that the 3C$^{pro}$-mediated cleavage activities have little if any effect on nucleocytoplasmic transport.

## 2A$^{pro}$, but not 3C$^{pro}$, mediates the selective shutdown of host translation

A hallmark of enterovirus infection is the selective shutdown of host cap-dependent translation initiation. 2A$^{pro}$ plays a major role in this shutdown by cleaving eIF4G, which is essential for 5'-cap dependent translation initiation but not for viral IRES-dependent translation initiation [27–31]. In fact, the C-terminal cleavage fragment of eIF4G has been implicated in enhancing vRNA translation [32–34]. Another cellular protein, death-associated protein 5 (DAP5), which is structurally homologous to the eIF4G C-terminal cleavage fragment, is suggested to drive viral translation initiation during the first round of translation before substantial eIF4G cleavage [34]. As infection progresses, 2A$^{pro}$ cleaves DAP5, resulting in a DAP5-N fragment that can still support IRES-mediated vRNA translation and a DAP5-C fragment that negatively affects cap-dependent translation [35]. Additionally, both 2A$^{pro}$ and 3C$^{pro}$ have been shown to cleave poly-(A)-binding protein (PABP), which is crucial for mRNA stability and translation initiation. However, the degree to which this contributes to the host shut-off is unclear. To examine the importance of 2A$^{pro}$ for the virus-induced host shut-off, we assessed eIF4G, DAP5, and PABP cleavage by WB at different time points post-infection. As expected, eIF4G cleavage products became apparent as early as 2 hpi in cells infected with wt 2A$^{pro}$ viruses. In contrast, the cellular eIF4G pool remained fully intact during 2Amut infection, even at later time points (Fig 4A). Similarly, DAP5 cleavage – detected from 6 hpi onward in CVB3–2Awt-infected cells – was absent in CVB3–2Amut-infected cells, even at later time points (Fig 4A). Cleavage of PABP, which we assessed using the monoclonal antibody 10E10 that only detects full-length PABP but not cleavage products, was observed in cells infected wt 2A$^{pro}$ viruses, but not in cells infected with CVB3–2Amut (Fig 4A), not even at 16 hpi (S3A Fig).

To evaluate the ability of CVB3 lacking 2A$^{pro}$ activity to induce host translation shut-off, we performed *in vivo* metabolic labeling using threonine-derived non-canonical amino acid tagging (THRONCAT) [36], a novel method that uses ethynylserine as a biorthogonal threonine analogue to mark newly synthesized proteins (Fig 4B) [36]. Infection of HeLa cells with CVB3–2Awt resulted in a progressive decrease in general protein synthesis starting at 3 hpi, with selective viral translation indicated by the appearance of specific bands representing viral proteins. Conversely, no such early selective translation shut-off was observed for CVB3–2Amut, but rather from 6 hpi a seemingly complete shut-off of all translation was observed, reminiscent of activation of the ISR (as exemplified by Arsenite treatment of cells, see lane 2). Notably, viral IRES-dependent translation has been suggested to be rendered independent of eIF2α (and thereby eIF2α phosphorylation) by 2A$^{pro}$, via a still unknown mechanism [37,38]. Due to the reduced replication efficiency of CVB3–2Amut and likely the inability to counter the ISR, viral proteins are poorly detectable in the background of cellular proteins. At later timepoints during infection, viral IRES-dependent translation decreases, as is reflected by the decrease in active translation we observed from 7hpi onwards. It is not known whether this is due to a switch from coupled translation and replication to assembly, or due to overall effects on cellular translation machinery due to the initiation of death programs in infected cells.

Activation of the ISR in enterovirus-infected cells is indicated by the phosphorylation of one of the ISR kinases, dsRNA-activated protein kinase (PKR), and the ensuing phosphorylation of alpha subunit of translation initiation factor

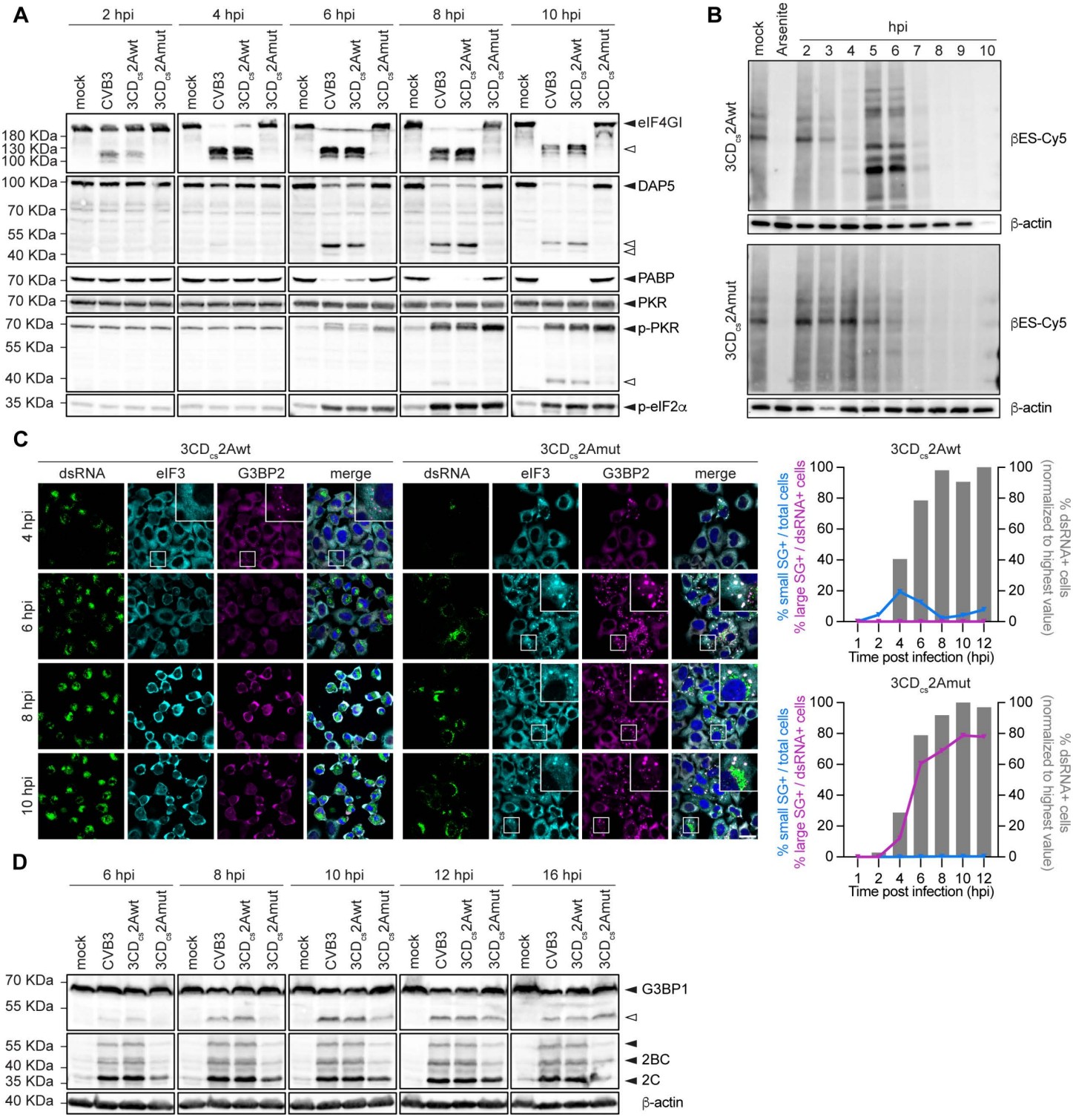

**Fig 4. 2A<sup>pro</sup>, but not 3C<sup>pro</sup>, selectively inhibits host translation and suppresses stress granule formation.** (A) Western-blot analysis of eIF4GI, DAP5, PABP, PKR, p-PKR and p-eIF2α. Cell lysates correspond to the same experiment shown in Fig 3A. Black triangles denote intact proteins, white triangles indicate cleavage products. (B) THRONCAT in vivo metabolic labeling of HeLa-R19 cells infected with 3 CD$_{cs}$-2Awt and 3 CD$_{cs}$-2Amut. Cells treated with sodium arsenite served as a positive control for translation arrest. Lysates were run on SDS-PAGE gels and subsequently transfered to

nitrocellulose membranes for analysis. (C) Representative images (left) and quantification (right) of SG formation in HeLa-R19 cells infected at an MOI of 10 with CVB3, 3 CD$_{cs}$-2Awt or 3 CD$_{cs}$-2Amut. Cells were fixed at 4, 6, 8, 10 hpi and analyzed by IF analysis for dsRNA as a marker of infection and eIF3 and G3BP2 as SG markers. Magnified regions are shown in white boxes. Scale bar: 25 μm. Quantification of small blue and large purple SG formation is shown on the right. At least 1000 cells were analyzed for each virus and time point. Small SG positivity is given as a percentage of total cells, since most cells are not yet positive for dsRNA at early timepoints. (D) Western blot analysis of G3BP1, CVB3 2C protein, and β-actin. HeLa-R19 cells, either mock-infected or infected with recombinant CVB3s at an MOI of 10, were lysed at 6, 8, 10, 12 and 16 hpi. Black triangles denote intact proteins, empty triangles indicate cleavage products.

eIF2 (eIF2α) [12,37,39–45]. Interestingly, we observed a similar pattern of PKR and eIF2α phosphorylation in CVB3–2Amut- and CVB3–2Awt-infected cells (Fig 4A). This observation is remarkable as, due to the delayed replication, there is considerably less dsRNA formed in CVB3–2Amut-infected cells. *We could detect cleavage fragments of PKR during both 3 CDcs-2Awt and 3 CDcs-2Amut infection (Fig 4A), reflecting cleavage of PKR by 3C$^{pro}$, a well-known phenomenon observed in enterovirus-infected cells* [43]. However, we were not able to detect any clear differences in the pool of full-length PKR, implying that only a small subset is cleaved. Moreover, this cleavage occurred only late in infection, after PKR activation and eIF2α phosphorylation, and thereby likely has little, if any, effect on ISR activation. Instead, our data suggest that 2A$^{pro}$ may delay PKR phosphorylation and ISR activation.

### Critical role of 2A$^{pro}$ in counteracting stress granule formation during enterovirus infection

Despite ISR activation, enteroviruses effectively suppress downstream canonical SG formation [12,46,47]. Initially, it was implied that 3C$^{pro}$ of both PV and EMCV, by cleaving G3BP1, a key SG nucleation factor, played a central role in this [48,49]. However, this has been challenged by the observation that 3C$^{pro}$, when expressed as a heterologous protein in the EMCV-Zn surrogate infection model, is unable to suppress SG formation, despite substantial G3BP1 cleavage. In contrast, 2A$^{pro}$ potently suppressed SG formation in the same system [47]. Yang and colleagues showed during EV-A71 infection that it is indeed 2A$^{pro}$ which renders the infected cells progressively more resistant to SG formation, and that this is conserved among different enterovirus species [12]. Of note, this was shown to rely on the catalytic activity of 2A$^{pro}$, rather than eIF4G cleavage specifically [50].

We performed IF analysis to assess SG formation during CVB3–2Amut infection, staining for the conventional stress granule markers G3BP2 and eIF3 (Fig 4C), as well as TIA-1 and Sam68 (S3B Fig). In line with earlier findings [12,46], CVB3–2Awt induces small SGs in the early phase of infection (4 hpi), which disappear at later time points [12,47]. These are thought to be the consequence of the 2A$^{pro}$-mediated cleavage of eIF4G and the ensuing inhibition of cap-dependent host mRNA translation. We find these early SGs to be positive for G3BP1 during CVB3 infection, whereas reports from other groups have observed them to be both positive in the case of CVB3 [46] and negative in the case of EV-A71 [12] for G3BP1. In line with the putative role of eIF4G cleavage, CVB3–2Amut did not induce the early appearance of small SGs. From 6 hpi onwards, however, CVB3–2Amut triggered the formation of large SGs containing eIF3, G3BP2 and TIA-1, but not Sam68.

Previously, both we and others have shown that these SGs are dependent on the PKR-mediated phosphorylation of eIF2α [12,47]. Importantly, albeit with considerably delayed kinetics, we still observe substantial cleavage of G3BP1 during CVB3–2Amut infection, consistent with continued 3C$^{pro}$ activity (Fig 4D). In conclusion, despite the cleavage of PKR and G3BP1 in CVB3–3 CD$_{cs}$-2Amut virus infected cells, SGs were not suppressed. Thus we confirm that CBV3 2A$^{pro}$ is critical for SG suppression during infection. However, we cannot exclude the possibility that the delayed kinetics of PKR and G3BP1 cleavage in CVB3–2Amut infection might shield a putative role of 3C$^{pro}$ in countering SGs during CVB3–2Awt infection.

## Differences in cell morphology and death between CVB3–2Awt and -2Amut infection

2A[pro] has been reported to cleave cytoskeletal components, such as cytokeratin 8 and dystrophin [51,52], and was shown to trigger apoptosis when individually expressed [53]. Therefore, we examined differences in cell morphology and cell death between cells infected with CVB3–2Awt and -2Amut using fluorescence microscopy (Fig 5A). Cells infected with CVB3–2Awt exhibited profound morphological changes, including cytoplasmic shrinkage, cytoskeleton rearrangement, indicated by β-tubulin staining, and altered nuclear shape. In contrast, cells infected with 2Amut virus showed morphological changes only at much later stages of infection, with apoptotic-like bodies and nuclear fragmentation appearing from 10 hpi onwards (Fig 5A and 5B). Importantly, neither of these was clearly observed during CVB3-wt infection. Furthermore, cells infected with CVB3–2Amut generally remained more attached and retained their stretched morphology throughout

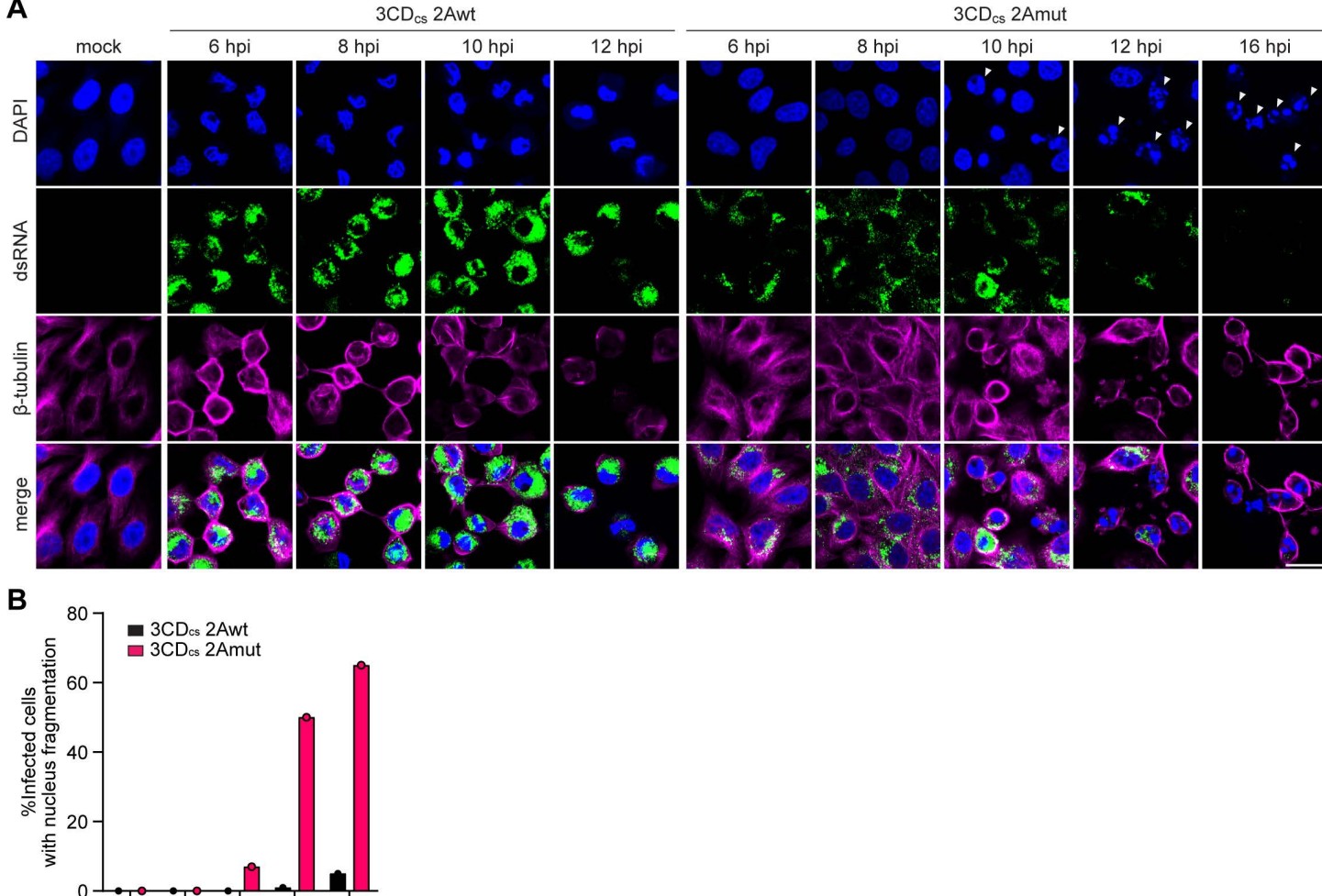

**Fig 5. Distinct effects of CVB3-2Awt and CVB3-2Amut on cell morphology and death.** (A) HeLa-R19 cells were infected with 3 CD$_{cs}$-2Awt and 3 CD$_{cs}$-2Amut at an MOI of 50, fixed at 6, 8, 10, 12 and 16 hpi and, and analyzed by IF analysis. Nuclei were stained with DAPI, dsRNA was detected as marker for infection, and β-tubulin was used as a cytoskeleton marker. Fragmented nuclei are indicated by white triangles. Scale bar: 20 μm. (B) Quantification of infected cells displaying nuclear fragmentation in panel (A).

infection, with cell shrinkage and rounding occurring only much later compared to cells infected with CVB3–2Awt, despite the presence of large amounts of viral proteins and RNA. This is in line with the marked absence of clear CPE observed in cells infected with CVB3–2Amut until very late in infection.

WB analysis revealed markers of apoptosis induction, namely caspase-7 (Casp7) and poly ADP-ribose polymerase (PARP) cleavage, in cells infected with CVB3, irrespective of 2Apro catalytic status (S4A Fig). Notably, we detected cleavage of Casp7 and PARP in CVB3–2Amut infected cells again only at much later time points. In conclusion, CVB3 infection seems to trigger apoptotic signaling, as shown by Casp7 and PARP cleavage. However, 2Apro catalytic activity appears essential to prevent the cellular morphological changes and nuclear fragmentation, normally associated with canonical apoptosis.

## 2Apro, but not 3Cpro, is essential to suppress type I IFN signaling

Enterovirus genome replication inevitably results in the formation of long dsRNA replication intermediates that are primarily recognized by the cytosolic RLR melanoma differentiation-associated protein 5 (MDA5) [54]. RIG-I, which recognizes dsRNA with 5'-triphosphate groups, which are absent on picornavirus genomic RNA, was found to be dispensable for sensing picornavirus infection in several studies [54–56], although a role of RIG-I in sensing EMCV and CVB3 infection in mouse macrophages has been suggested [57]. Extensive work has shown that both 2Apro and 3Cpro cleave key components of the RLR pathway. 2Apro cleaves MDA5 and MAVS, whereas 3Cpro targets RIG-I and MAVS [58–63]. However, in an EMCV-L(Zn) surrogate infection model, 2Apro, rather than 3Cpro, mediated effective suppression of IFN-β responses [47].

To evaluate the respective roles of 2Apro and 3Cpro in suppressing the RLR signaling pathway and IFN-α/β interferon production during CVB3 infection, we conducted qPCR experiments to measure mRNA levels of IFN-β and interferon-induced protein with tetratricopeptide repeats 1 (IFIT1), an interferon-stimulated gene (ISG) product that can also be induced directly by phosphorylated IRF3.

In line with the results shown in (Fig 1), we observed a decrease in viral RNA copies during infection with CVB3–2Amut (Fig 6A). Despite this slower replication, and in contrast to CVB3–2Awt, infection with CVB3–2Amut resulted in significant induction of IFN-β and IFIT1 gene transcription, from 6 hpi onwards (Fig 6A). To gain more insight into the underlying mechanism, we performed WB analysis to examine the cleavage of MDA5, RIG-I and MAVS, alongside phosphorylation of the downstream transcription factor IRF3. We observed cleavage of MDA5, MAVS, and RIG-I from 6 hpi onwards during CVB3–2Awt infection (Fig 6B). Importantly, we detected no IRF3 phosphorylation, even at later time points. In contrast, no cleavage of MDA5 and MAVS was observed in cells infected with CVB3–2Amut up to 10 hpi. Only at later time points, a MAVS cleavage fragment was observed of similar size as previously reported upon cleavage by 3Cpro (S5A Fig). This implies that either 3Cpro-mediated cleavage of MAVS is relatively inefficient and only occurs very late in infection, or that initial 2Apro-mediated cleavage is required to facilitate further proteolytic cleavage by 3Cpro. We still detected cleavage of RIG-I during CVB3–2Amut infection, albeit at later time points (i.e., from 8 hpi onwards). Despite this, IRF3 phosphorylation could be detected from 8 hpi onwards, consistent with transcriptional activation of the IFN response observed earlier. These data provide evidence that 2Apro is the main antagonist of the IFN-α/β response during CVB3 infection.

Aside from its central role in suppressing RLR signaling and subsequent IFN production, 2Apro may also be important for enteroviruses to overcome the antiviral state established in cells via paracrine IFN-α/β signaling [64]. Previously, it was shown that the addition of the poliovirus 2Apro gene to the EMCV genome allowed EMCV to replicate in IFN-alpha-pretreated cells [64]. To assess if CVB3, in the absence of 2Apro activity, is unable to overcome IFN pretreatment, we performed luciferase assays on cells pretreated with IFN-α2, infected with renilla-luciferase (RLuc) variants of CVB3–2Awt and -2Amut. We observed that whereas CVB3–2Awt replication remained largely unaffected, CVB3–2Amut was highly sensitive to pretreatment with IFN-α2, with progressively more inhibition at higher concentrations (Fig 6C). This confirms that 2Apro is critical to overcome the antiviral state induced by IFN signaling.

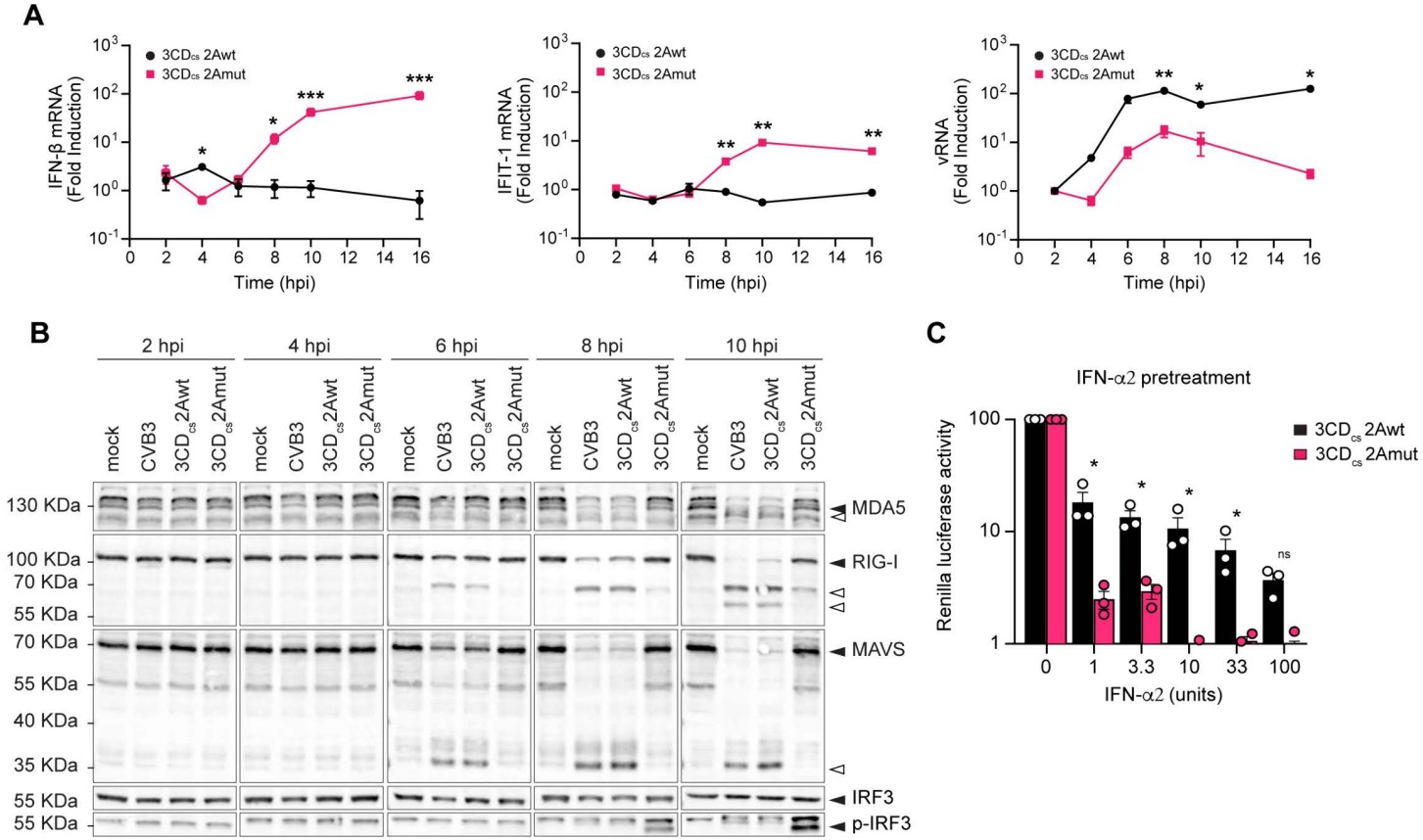

**Fig 6. 2A$^{pro}$ is the key viral protease responsible for the suppression of type I interferon signaling.** (A) HeLa-R19 cells were infected at an MOI of 10 and lysed at 2, 4, 6, 8, 10 and 16 hpi. Total RNA was analyzed by RT-qPCR for IFN-β, IFIT1, viral RNA (vRNA) and actin. IFN-β, IFIT1 and vRNA levels were calculated as fold induction compared to levels in mock-infected cells, after correction for actin mRNA levels, and normalized to the expression levels at 2 hpi. Data represent mean±SEM of three technical replicates. Statistical significance was calculated by multiple two-tailed unpaired t-tests; * p<0.05; ** p<0.01; *** p<0.001. (B) Western blot analysis for MDA5, RIG-I, MAVS, IF3 and p-IRF3. Cell lysates belong to the same experiment shown in Fig 3A. Black triangles point out intact proteins, white triangles indicate the cleavage products. (C) Hela-R19 cells were pretreated for 16 hrs with IFN-α and subsequently infected with RLuc-3 CD$_{cs}$-2Awt and -2Amut viruses. Cells were lysed at 6 hpi and luciferase levels were assessed. Data represent the mean±SEM of three independent experiments normalized to the untreated control. Statistical significance was assessed by multiple two-tailed unpaired t-tests with multiple comparisons corrections using the Bonferroni-Dunn method (* p<0.05; ** p<0.01; *** p<0.001).

## Single-molecule imaging to elucidate the dynamics of CVB3–2Amut translation and replication

To gain more insight into the delay in replication that we observed for CVB3–2Amut, we turned to a single cell imaging assay recently developed in our lab termed <u>VI</u>rus <u>I</u>nfection <u>R</u>eal-time <u>IM</u>aging (VIRIM) [14]. Briefly, VIRIM is a two-component system consisting of cells expressing a single-chain variable antibody fragment fused to a fluorescent protein (SunTag-antibody, STAb-FP) and viruses that have been genetically engineered to express a short suntag peptide array at the N-terminus of the viral polyprotein followed by a 3C$^{pro}$ cleavage site (Fig 7A). Upon translation of the viral genome in STAb-FP expressing cells infected with Suntag-CVB3 (ST-CVB3), STAb-FP will co-translationally bind to the suntag peptides emerging from the ribosome (Fig 7B). Since multiple ribosomes simultaneously translate a single vRNA, many STAb-FPs will be recruited to vRNAs undergoing translation, and these translating vRNAs can be observed as bright fluorescent foci using spinning-disk confocal microscopy. Importantly, cessation of translation results in the release of individual 'mature', STAb-FP-bound SunTag peptides with their associated fluorescence from the vRNA. Such mature SunTag

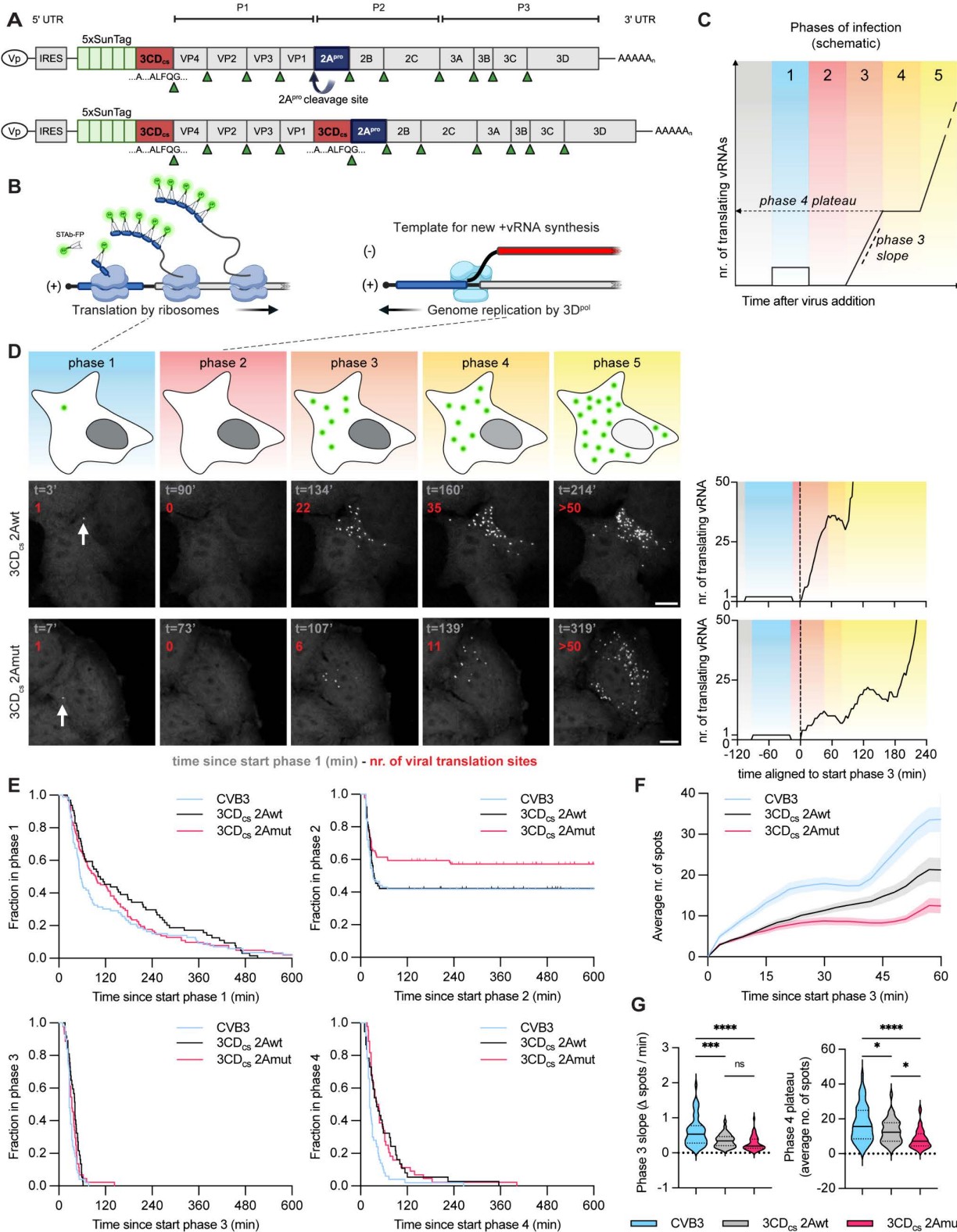

**Fig 7. Single-molecule imaging to dissect the role of 2A^pro during the earliest phases of CVB3 infection.** (A) Schematic representation of the SunTag-CVB3 viruses. 5xSunTag cassette flanked at its 3'-side by a 3 CD_cs was inserted before the P1 region of CVB3, 3 CD_cs-2Awt, or 3 CD_cs-2Amut.

(B) Schematic representation of the VIRIM assay. Left panel shows active translation. Right panel denotes situation during viral replication. The blue region of the viral genome indicates the SunTag peptide sequences. Note the opposing directions of both processes, and their consequent temporal exclusiveness. (C) Schematic representation of the different phases of early enterovirus infection as defined by Boersma et al. 2020. (D) Pictures of the different phases of viral infection, as observed for representative infections with 3 CD$_{cs}$-2Awt, and 3 CD$_{cs}$-2Amut, respectively. The time since start phase 1 is stated in grey. The numer of translating vRNAs in the infected cell is stated in red. Schematic illustration of VIRIM spot number is shown in the top panel. The far right graphs show VIRIM spot count for the infections shown in the representative images on the left. (E) Kaplan-Meier curves depicting the duration of the different phases of infection for CVB3, 3 CD$_{cs}$-2Awt, and 3 CD$_{cs}$-2Amut. (F) The average number of spots observed over time during CVB3, 3 CD$_{cs}$-2Awt, and 3 CD$_{cs}$-2Amut infection. Shaded areas represent SEM. (G) Violin plots depicting the average rate of increase in the number of translating vRNAs during phase 3 (phase 3 slope) (left), and the average number of translating vRNAs in phase 4 (phase 4 plateau, see also (C)) (right). Scale bar 10 μm. The illustration in (A, B, and D) were (partially) created in BioRender. Schipper, J. (2025) https://BioRender.com/unbzo59.

peptides are subsequently dispersed throughout the cytoplasm, where they are too dim to be observed as fluorescent foci. Therefore, fluorescent foci specifically mark sites of active translation. As vRNA translation and replication occur in opposite directions on the same vRNA strand, they are temporally exclusive. Consequently, translation necessarily ceases upon the start of replication, leading to a disappearance of VIRIM signal (Fig 7B). Thus, by analyzing the number of VIRIM foci over time in a single infected cell, we can identify distinct phases of translation and replication of CVB3 during early infection (Fig 7C and 7D) [14].

We engineered CVB3–2Awt and -2Amut to encode a 5x-SunTag (ST) array before VP4. Recovered viruses stably maintained the insert over passage. During time-lapse imaging of infected SunTag cells additionally expressing a blue flu-orescent protein- (BFP-)NLS NCTD reporter construct (as described previously [14]), we observed leakage of the reporter construct from the nucleus for ST-CVB3–2Awt, starting within minutes and being complete within the first 2h after initial translation was detected (S6A Fig). Consistent with our earlier data, BFP-NLS remained confined to the nucleus in cells infected with ST-CVB3–2Amut, even at late times post infection (S6A Fig). We could discern the same general phases for both ST-CVB3–2Awt and -2Amut as we previously observed for ST-CVB3-wt infections (Fig 7E) [14]. We observed the appearance of a single spot upon translation of the incoming vRNA (phase 1), which disappeared as result of cessation of translation and the start of replication (phase 2), after which new foci appeared due to translation of newly synthesized vRNAs (phase 3), followed by translation of newly-synthesized vRNAs (phase 4) and finally and second wave of repli-cation and translation that results in a further increase in VIRIM foci (phase 5). The duration of phases was similar for ST-CVB3–2Awt and -2Amut infections, with no apparent differences in timing of either translation or replication (Fig 7E). However, we observed significantly less VIRIM foci reappearing in phase 3 and beyond during ST-CVB3–2Amut infec-tion (Fig 7F and 7G), implying replication was less efficient. Moreover, a substantially higher fraction of infections were arrested during initial replication of ST-CVB3–2Amut (Fig 7E). This supports our earlier observations that initial replication constitutes the most vulnerable phase of infection [14]. Moreover, it illustrates the potency and importance of 2A$^{pro}$ cat-alytic activity to facilitate the very first round of replication. Of note, we observed a somewhat higher phase 2 arrest rate for ST-CVB3-wt compared to what we observed previously, i.e., ~20–25% [14], which is likely due to the use of another monoclonal cell line in the current study. In accordance with previous results, we observed that the duration of phase 1 (initial translation), was somewhat extended in cells with unsuccessful replication. Importantly, this held true regardless of 2A$^{pro}$ catalytic activity (S6C Fig). We previously observed that for a fraction of infections during which the first round of replication is not successful, viral translation reinitiates on the incoming vRNA, after which there is another attempt at replication. These translation pulses and replication breaks can occur multiple times, and can either culminate in replica-tion success or arrest [14]. Whereas we previously observed that the number of pulses and breaks increases starkly upon disruption of viral replication (with a 3D$^{pol}$ inhibitor), no such increase was observed for ST-CVB3–2Amut (S6D Fig).

## 2A$^{pro}$ progressively stimulates viral translation efficiency during the course of infection

VIRIM can also be used to assess viral translation efficiencies. The C-terminal cleavage product of eIF4G stimulates enteroviral IRES-mediated translation [32–34]. Additionally, nuclear-resident RNA-binding proteins, which are relocalized

to the cytosol due to the NCTD, have also been proposed to stimulate enteroviral translation [6]. Therefore, we set out to investigate whether, because of the absence of 2A^pro-mediated eIF4G cleavage and NCTD, viral translation would be progressively less efficient over the course of CVB3–2Amut infection. VIRIM translation spot intensity is dictated by the number of SunTag-containing nascent chains associated to ribosomes that are translating a vRNA, and hence by the efficiency of viral translation initiation (S7A Fig). Thus, to compare translation efficiency over time, we measured the intensity of VIRIM translation foci within single cells infected with either ST-CVB3–2Awt or -2Amut. During ST-CVB3–2Awt infection, we observed that spot intensity on average increased from phase 1 to phase 4. However, no such significant increase could be observed for 2Amut infections (Fig 8A). These results are in line with a progressive increase in translation efficiency over the course of infection with CVB3 carrying a proteolytically active 2A^pro, brought about by the diverse and critical actions of 2A^pro, and might (partially) underlie the decrease in the efficacy of replication in ST-CVB3–2Amut infected cells. Importantly, during phase 1, when no significant cleavage of eIF4G has yet occurred, there were no significant differences in translation efficiency between ST-CVB3–2Awt and ST-CVB3–2Amut (Fig 8B). The higher rate of phase 2 arrest is therefore unlikely the consequence of a lower translation efficiency during phase 1, but rather suggests a direct role of 2A^pro in the first round of viral replication.

## Discussion

Due to the constrained coding capacity of viral genomes, viruses have often evolved multiple functions within single proteins, while simultaneously exhibiting functional redundancy and overlap with other viral proteins. Moreover, they regularly function interdependently, engaging both other viral proteins and host factors to exert their role. Thus, it can be challenging to dissect the respective importance of viral factors to observed effects, and proper context truly matters. In this regard, we have been similarly constrained in our understanding of the critical security protein 2A^pro. Due to the essential nature of its role in polyprotein processing, we have so far relied mostly on *in vitro* work, transient overexpression experiments, and surrogate infection models to study the functions of 2A^pro. Although these gave key insights, they do not enable us to fully assess its function within the proper infection context. Moreover, the respective importance of the other

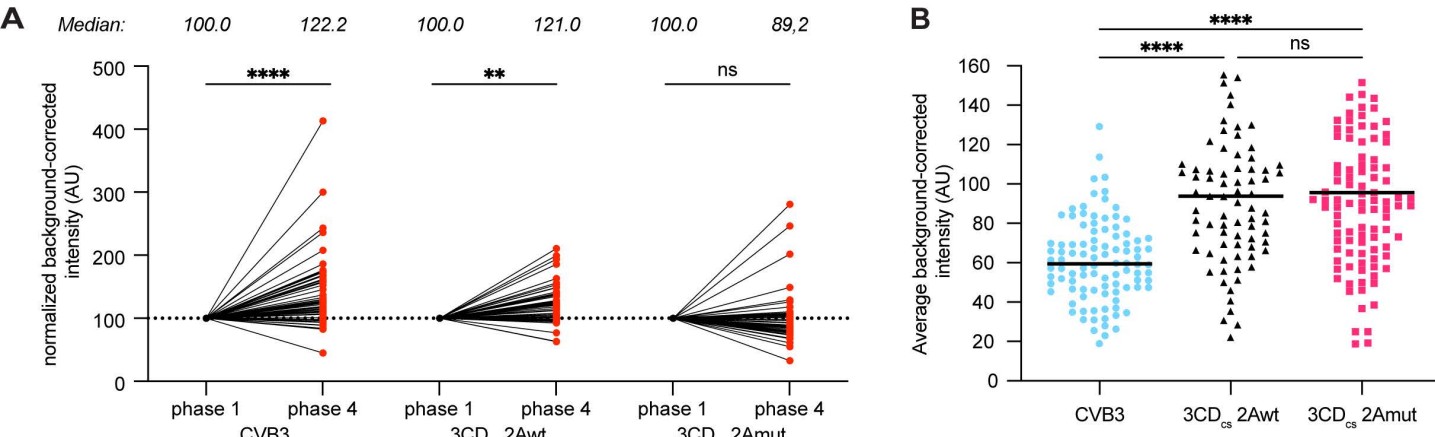

**Fig 8. The increase in viral translation efficiency during infection depends on the action of 2A^pro.** (A) VIRIM spot intensity was measured during phase 1 and phase 4 in the same infected cells. Spot intensities are plotted relative to phase 1 spot intensity. Each line and red dot represents an individual infected cell. Median values are indicated above the graphs. (B) Normalized intensity (AU) of phase 1 translation spots during infection with CVB3, 3 CD_cs-2Awt, and 3 CD_cs-2Amut. The higher intensity of the 3 CD_cs viruses can be explained by the comparatively longer effective transcript length (i.e., the length of vRNA that is decoded by the ribosome before the SunTag containing nascent polypeptide is released from the ribosome) of these viruses due to the replacement of the 2A_cs with the 3 CD_cs. Statistical significance was assessed by a two-way analysis of variance (ANOVA) with multiple comparisons testing. * $p < 0.05$; ** $p < 0.01$; *** $p < 0.001$.

enterovirus protease, 3C$^{pro}$, towards many of the observed changes in the cell during infection, has remained convoluted. Here, we constructed viable CVB3 viruses with catalytically inactive 2A$^{pro}$. We demonstrate that CVB3–2Amut viruses are hampered in their replication to various degrees in different cell lines and that these viruses are severely attenuated in a mouse model. We establish a critical role for 2A$^{pro}$, and not 3C$^{pro}$, during CVB3 infection in the rapid induction of NCTD via the cleavage of nucleoporins, mediating a selective translation shutdown by the cleavage of eIF4GI and PABP, suppressing SG formation, and cleaving RLR pathway proteins, inhibiting IRF3 activation and suppressing IFN-α/β gene transcription (for an overview see S8 Fig). Furthermore, we demonstrate that 2A$^{pro}$ is critical for overcoming early restriction by ISGs. Finally, making use of an advanced single-molecule live cell imaging approach, we reveal that 2A$^{pro}$ is important for the initial round of replication. In conclusion, we establish with a wide array of techniques and approaches that during CVB3 infection 2A$^{pro}$, in contrast to 3C$^{pro}$, plays a critical role in subverting antiviral responses and in establishing a favorable host environment to expedite viral replication. As such, 2A$^{pro}$ constitutes a true "security protein"; highly important for virulence and fitness, but, at least in cell culture, dispensable for viral growth [13,65].

Our results raise questions about the proposed roles of 3C$^{pro}$ in the processes described. Our data support previous suggestions that cleavage of Nup98 by 2A$^{pro}$, which occurs early already upon translation of the incoming viral RNA genome and is exclusively mediated by 2A$^{pro}$, is essential for triggering the NCTD. Cleavage of other nucleoporins by 3C$^{pro}$ late in infection may have some subtle contributions to the disruption of nuclear pore integrity in infected cells, but the relevance hereof for the viral life cycle is unknown. Furthermore, our data show that eIF4G is rapidly and exclusively cleaved by 2A$^{pro}$ early in infection, leading to inhibition of cap-dependent host mRNA translation. PABP, is also cleaved during infection, reportedly by both PV 2A$^{pro}$ and 3C$^{pro}$ [66–68]. During CVB3–2Amut infection however, no reduction in PABP levels was observed, although it cannot be excluded that 3C$^{pro}$ cleaves a small subset of cellular ribosome-associated PABP specifically, as suggested previously [67]. The lack of efficient PABP cleavage by 3C$^{pro}$ may be explained by the comparatively low efficiency of total PABP cleavage by 3C$^{pro}$ observed *in vitro* [66]. Overall, our data thus suggest that 2A$^{pro}$ is exclusively involved in shutting off host mRNA translation by cleaving eIF4G and PABP.

Our results suggest that CVB3 2A$^{pro}$ is responsible for suppression of canonical SGs later in infection, in line with results obtained for EV-A71 2A$^{pro}$ [12]. The viral disruption of late SGs was originally attributed to the ability of 3C$^{pro}$ to cleave G3BP1, as mutation of the 3C$^{pro}$ cleavage site in G3BP1 resulted in a reduced ability to disrupt SGs, and a decrease in PV replication [48]. Strikingly, we observed fulminant SG formation, despite substantial G3BP1 cleavage during CVB3–2Amut infection. This could also imply a potential novel role for 2A$^{pro}$, in facilitating efficient cleavage of G3BP1 via an unknown mechanism. Alternatively, the reduced ability to disrupt SGs in cells expressing 3C$^{pro}$ cleavage-resistant-G3BP1 was due to reduced replication [48]. It remains unclear how 2A$^{pro}$ inhibits SG formation. Yang and colleagues demonstrated that 2A$^{pro}$ abrogates an RNA-dependent interaction between G3BP1 and eIF4GI, which is critical to SG formation [50]. However, it is challenging to separate cause from consequence, as the observed interaction might be consequential rather than the driving factor behind SG formation. Interestingly, the L protein of EMCV, which similarly suppresses formation of canonical SGs late in infection, also abrogated the interaction between eIF4GI and G3BP1, although this protein is not a protease and thus cannot cleave eIF4GI [50]. Alternatively, the ability of 2A$^{pro}$ to disrupt SGs might relate to its reported cleavage of Tudor-Domain Containing 3 (TDRD3), a nuclear-resident protein which is recruited to SGs, and which has been suggested to stabilize SGs [69,70]. Taken together, a detailed understanding of the mechanism underlying SG inhibition by 2A$^{pro}$ remains to be established.

Enteroviruses efficiently suppress IRF3 activation to prevent IFN-α/β gene transcription. Cleavage of both RLRs and MAVS by 2A$^{pro}$ and 3C$^{pro}$ has been implicated in this inhibition. A role of 3C$^{pro}$ in cleaving MAVS was proposed based on ectopic overexpression of this protease [60]. Other studies, however, showed a prominent role of 2A$^{pro}$, but not 3C$^{pro}$, in MAVS cleavage [61,62]. Consistently, we observed 3C$^{pro}$ cleavage products of MAVS in CVB3-infected cells only at very late time points (>12 hpi), whereas the main MAVS cleavage, perpetrated by 2A$^{pro}$, was observed much earlier (>6 hpi), simultaneous with cleavage of MDA5 by 2A$^{pro}$. Notably, initial transcriptional activation of IFN in CVB3–2Amut infected cells

starts already at the same time point, which implies that the sensing of dsRNA has occurred before any substantial cleavage of MDA5, MAVS, or RIG-I. Granted that our ability to detect cleavage events might not be perfect, this still begs the question of how, temporally speaking, 2A$^{pro}$-mediated cleavage of MDA5 and MAVS can counteract IFN gene transcription so efficiently. Possibly, other 2A$^{pro}$-dependent mechanisms, such as the NCTD, contribute to this inhibition. SGs have previously been shown to serve as platforms for RLR signaling [47,49,71], but knockout of both G3BP1 and -2, which completely abrogates the ability of Hela cells to form SGs, resulted in only a minor reduction in IFN production during EMCV-L(Zn) infection. Hence, the 2A$^{pro}$-mediated inhibition of SGs fails to account for the full extent of IFN suppression. EMCV-wt efficiently suppresses IRF3 activation and IFN gene transcription through its L protein, even though RLRs and MAVS are not cleaved. Thus far, the only known activity of L is to disturb nucleocytoplasmic trafficking by recruiting ribosomal s6 kinases to the nuclear envelope where they cause hyperphosphorylation of FG-nucleoporins, such as Nup98 [6]. Although the correlation between the NCTD and the suppression of IRF3 activation is poorly understood, it raises the possibility that, by analogy, the activity of 2A$^{pro}$ to cleave Nup98 may underly its ability to rapidly block activation of IRF3. Further research is needed to fully unravel the mechanistic underpinnings of interferon inhibition by 2A$^{pro}$ during enterovirus infection.

The construction and characterization of CVB3 with catalytically inactive 2A$^{pro}$ also revealed important novel clues regarding the role of 2A$^{pro}$ in several other processes during CVB3 infection. For instance, we observed that CVB3–2Amut infected cells retained their regular morphology much longer, in line with development of a clear CPE only at very late time points. This might be partly explained by the lack of 2A$^{pro}$-mediated cleavage of cytoskeleton proteins. Strikingly, we observed much less nuclear deformation – a hallmark of enterovirus infection - but were able to detect fragmentation of nuclear DNA. It is tempting to speculate that this might relate to faulty trafficking and mislocalization of the main DNA fragmentation factor (DFFB) due to the NCTD during CVB3–2Awt infection [72]. Furthermore, we found evidence for a function of 2A$^{pro}$ to modulate PKR activation. Both PKR and eIF2α become phosphorylated during CVB3 infection. Remarkably, despite the general delay in replication observed for CVB3–2Amut, phosphorylation of PKR and eIF2α seems to manifest itself in a similar pattern as during infection with CVB3–2Awt. This implies that CVB3 2A$^{pro}$ proteolytic activity delays the activation of PKR and downstream eIF2α phosphorylation, in line with a previous observation with EV-A71 [12]. The importance of this function remains to be established as enteroviral IRES-mediated translation has been shown to be relatively insensitive to eIF2α phosphorylation [37,38]. It is noteworthy that we did not find evidence for a role of 2A$^{pro}$ in modulating PKR and eIF2α phosphorylation when expressed as a heterologous protein in a surrogate EMCV infection model [47], emphasizing the importance of the proper infection context to gain insight into the function(s) of viral proteins.

To gain more insight into the role of 2A$^{pro}$ during the earliest phases of infection, we resorted to an advanced single-molecule imaging approach called VIRIM. We show that the efficiency of replication during early CVB3–2Amut infection is markedly lower, illustrated by the slower increase in the number of translating vRNA as infection progresses. Moreover, we show that in the absence of 2A$^{pro}$ activity, viral translation efficiency, as measured by VIRIM spot intensity, does not increase as infection progresses. It is conceivable that the lack of accessibility to proviral host factors, which normally relocalize from the nucleus upon the induction of NCTD to promote translation and replication, combined with the absence of cleaved eIF4G and other host factors thought to promote viral translation, leads to reduced efficiency in translation and replication. Distinct from the effect on the efficiency of replication in late-stage infection, we also show that the initial replication of the incoming vRNA is more vulnerable in the absence of 2A$^{pro}$, as is evidenced by the higher phase 2 arrest rate. Importantly, this is unlikely due to a decrease in the amount of viral proteins that are synthesized and available for initial replication, as the efficiency of translation on the incoming vRNA during the preceding phase 1 is similar for CVB3–2Awt and -2Amut. Rather, the higher rate of failure for the first round of replication might relate to the inaccessibility of replication factors that are normally relocated from nucleus. Alternatively, it might be due to decreased RNA stability as Kempf and Barton have shown that 2A$^{pro}$ catalytic activity was important for viral RNA stability and to suppress host mRNA turnover machinery [73].

We previously showed that replication of incoming vRNAs is a key point at which ISGs act to restrict viral infection [14]. Here, we show that 2A$^{pro}$ is important to overcome the antiviral state brought about by ISGs, in line with previous work

[64], although the exact mechanism remains unclear. It remains to be established whether there is overlap in the identity of the putative restriction factors that are involved in the increased phase 2 arrest rate observed for CVB3–2Amut and the IFN-stimulated restriction factors that interfere with vRNA replication. Variations in the steady-state expression levels of such restriction factors among different cell lines may well explain the major differences that we observed in their susceptibility for efficient infection with CVB3–2Amut (Fig 1E).

## Limitations

Here, we thoroughly examined the respective contributions of the viral proteases 2A$^{pro}$ and 3C$^{pro}$, within the proper infection context, to many of the processes underlying effective enterovirus replication and virulence. Notwithstanding that, our study has certain limitations, particularly as it relates to the relative importance of 3C$^{pro}$. First, although we directly study many of the functions of 2A$^{pro}$ during infection, most of our conclusions relating to putative roles of 3C$^{pro}$ are based on indirect evidence. Ideally, one would similarly engineer enteroviruses with proteolytically-inactive 3C$^{pro}$, but this is precluded by the sheer number of autoproteolytic cleavage events for which 3C$^{pro}$ is essential during polyprotein processing. Second, we do not find a clear role of 3C$^{pro}$ for certain processes in which it was previously implicated. One might argue that the absence of such effects could relate to the relatively low expression level of 3C$^{pro}$ during CVB3–2Amut infection due to less efficient replication. However, the observation of other 3C$^{pro}$ cleavage events, such as those of PKR, G3BP1, and RIG-I, argues against such a notion. Third, Even though 2A$^{pro}$ is catalytically inactive during CVB3–2Amut infection, the protein is still made, and thus we cannot exclude that it might exert some dominant-negative effects on both viral- and host processes. For example, previous studies successfully used catalytically inactive 2A$^{pro}$ for pull-down studies [21], demonstrating its continued ability to bind substrates, potentially affecting their function. Fourth, we mostly made use of HeLa cells as a model system, because CVB3–2Amut replicated relatively well in this cell line, giving us a good window for assessing the role of 2A$^{pro}$ in many processes. In other cell lines, such as A549, CVB3–2Amut replication was much more compromised, the reason for which is unknown. Possibly, this might relate to higher steady-state expression of specific restriction factors that need to be countered by 2A$^{pro}$, but this requires further study. Finally, our study on the importance of 2A$^{pro}$ was restricted on CVB3, which is a commonly used model enterovirus. However, we argue that – based on the evolutionary conserved functions of the enterovirus proteases [47,74] – our results have broad implications for a better understanding of the interplay of other enteroviruses with host cells as well.

## Methods

### Ethics statement

Housing conditions and experimental procedures were approved by the ethics committee of animal experimentation of KU Leuven (license P179/2023).

**Cell lines.** HEK293-T (ATCC CRL-3216), HeLa-R19 (RRID: CVCL_M763), BGM (RRID: CVCL_4125), Vero E6 (ATCC CRL-1586), A549 (ATCC CCL-185), Hela-R19 saGFP-STAb BFP-NLS STAb (derived from Hela-R19, made as described previously in Boersma 2020, Bruurs 2025) were maintained in Dulbecco's Modified Eagle's Medium (DMEM) supplemented with 10% (V/V) fetal bovine serum (FBS) and 100 units/ml penicillin and streptomycin (P/S).

**Virus design and production.** Recombinant CVB3 viruses were derived from the pRibCVB3/T7 plasmid, which contains the cDNA of CVB3 strain Nancy driven by a T7 RNA polymerase promoter (Wessels et al., 2006 [75]). Mutations in the 2A$^{pro}$ sequence (H21A, D39A, and C110A) were introduced by site-directed mutagenesis of the pRib infectious clone. The 2A$^{pro}$ cleavage site between the P1-P2 region was replaced by: (I) the IRES sequence of Theiler's encephalomyelitis virus (TMEV) genome, (II) a self-cleaving peptide from *Thosea asigna* virus (T2A), or (III) a sequence corresponding to the 3C/3D cleavage site of CVB3 recognized by 3C$^{pro}$ (3 CD$_{cs}$). 5x-Suntag-CVB3 viruses were generated as described earlier (Boersma et al., 2020 [14]). Briefly, we introduced a 5x-SunTag cassette with a 3 CD cleavage site upstream of

the VP4 region of the main viral ORF in the infectious cDNA clone plasmids of CVB3-wt, CVB3–3 CDcs-2Awt, or CVB3–3 CDcs-2Amut. *Renilla* luciferase (RLuc)-CVB3 viruses were generated by introducing the *Renilla* luciferase gene upstream of the VP4 gene separated by the 3C/3D cleavage site recognized by 3C$^{pro}$. Virus stocks were generated by transfecting purified *in vitro* transcribed viral RNA (RiboMAX, Promega) in HEK293-T cells. After 2–4 days, when total cytopathic effect (CPE) was visible, the supernatant containing virus was subjected to three freeze-thaw cycles, cell debris was pelleted at 4,000x$g$ for 15 minutes and the virus was concentrated from the supernatants by ultracentrifugation through a 30% sucrose cushion at 140,000x$g$ for 16 hrs at 4$^o$C in a SW32Ti rotor. Virus pellets were resuspended in phosphate-buffer saline (PBS), aliquoted, and stored at -80$^o$C. The viruses were characterized by isolating viral RNA from 150 μl aliquots of the cell culture supernatant with the NucleoSpin RNA Virus kit (Macherey-Nagel) followed by conventional RT-PCR and bidirectional Sanger sequence analysis of the inserted (I) IRES, (II) T2A, or (III) 3 CDcs sequence between P1-P2 region, 2A$^{pro}$ and 5x-SunTag array sequence. Viral titers, determined on HeLa cells by endpoint titration and calculated by the Spearman-Kaerber formula, are averaged from three independent experiments.

**Mice experiment Materials and methods.** SJL/JRj mice (6 weeks old) were purchased from Janvier Labs (France) and were infected intraperitoneally with equivalent viral doses (either $1.0*10^7$ TCID$_{50}$/ml or $5.0*10^5$ TCID$_{50}$/ml) of CVB3, CVB3–2Awt or CVB3–2Amut. At day 3 post-infection (pi), mice were euthanized by intraperitoneal (IP) injection of 100 μL Dolethal (200 mg/mL sodium pentobarbital, Vétoquinol SA) and pieces of pancreas were collected for quantification of viral RNA, infectious virus titers by RT-qPCR, and end-point virus titration, respectively. In addition, other pieces of pancreas were fixed in 4% formaldehyde for histopathological analysis.

Pancreatic histopatology was assessed and scored as follows. For necrosis: 0 = tissue unaffected, 1 = < 10% of tissue affected, 2 = 11–25% of tissue affected, 3; 26–50% of tissue affected, 4 = more than 50% of tissue affected; for inflammation and edema: 0 = none, 1 = mild, 2 = moderate, 3 = severe. Analyses were conducted blinded to the experimental groups. For each mouse, 10 field of views obtained with a 20x objective were analyzed and the median scores determined.

**Growth curve and quantitative PCR (RT-qPCR).** HeLa R19 cells were seeded in flat-bottom 96-well plates ($10^4$ cells/well) and after 16 hrs recovery were infected with the recombinant CVB3 at MOI = 5 TCID$_{50}$/ml for 30 min. At indicated time points post-infection plates were either freeze-thawed three times to lyse the cells for titration or lysed using RNA lysis buffer for isolation of total RNA (Machery-Nagel) for RT-qPCR. Reverse transcription was set up using TaqMan Reverse Transcription Reagents (ThermoFisher) and qPCR analysis was performed using specific primers for IFN-β (5'-ATGA CCAACAAGTGTCTCCTCC-3' and 5'-GCTCATGGAAAGAGCTGTAGTG-3'), IFIT1 (5'-TGCTCCAGACTATCCT TGACCT-3' and 5'-TCTCAGAGGAGCCTGGCTAA-3'), β-actin (5'-CCTTCCTGGGCATGGAGTCCTG-3' and 5'-GGA GCAATGATCTTGATCTTC-3'), and CVB3 viral RNA (5'-CGTGGGGGCTACAATCAAGTT-3' and 5'-TAACAGGAGCTTTG GGCATC-3') using the Fast SYBR green Master Mix (ThermoFisher). Relative levels of IFIT1, IFN-β mRNA and vRNA were normalized to actin expression. In each experiment, expression was determined from three technical replicates.

**Luciferase assay.** HeLa R19 cells ($10^4$ cells/well) and HEK293-T cells (1.2x10 cells/well) were seeded in 96-well plates. After 16-hour recovery, cells were transfected with *in vitro* transcribed RLuc-CVB3 RNA using Lipofectamine2000 (Invitrogen). At indicated time point post infection, cells were washed with PBS and lysed in *Renilla* luciferase assay lysis buffer (Promega). Luciferase activity was measured using a *Renilla* luciferase assay system (Promega) with a GloMax Explorer luminometer (Promega). Luminescence was measured as relative light units per second (RLU/s). For the interferon (IFN) pretreatment experiment, HeLa cells were treated with the indicated concentrations of IFN-α2 (Sigma-Aldrich) for 16 hrs. Cells were infected with RLuc-CVB3 recombinant viruses at MOI of 0.1 TCID$_{50}$/ml for 30 min. Following infection, the virus inoculum was removed and replaced with complete medium containing IFN-α2 at the same concentrations used during the pretreatment phase. Luciferase activity was measured as described above.

**Antibodies.** The following antibodies were used for WB analysis: rabbit anti-eIF4GI (1:1000, Cell Signaling, #2498), rabbit anti-DAP5 (1:1000, Invitrogen, #PA-21377), mouse anti-PABP (1:1000, SantaCruz, #sc-32318), anti-PKR (1:4000, BD Biosciences, #BDB610764), rabbit anti-PKR-p T446 (1:1000, Abcam, #ab32036), rabbit anti-eIF2α-p (1:1000, Abcam,

#32157), rabbit anti-Nup98 (1:1000, Cell Signaling, #2598), rabbit anti-G3BP1 (1:1000, Aviva, #ARP37713-T100), rabbit anti-MDA5 (1:5000, kindly provided by Paola Barral), rabbit anti-RIG-I (1:500, Abgent, #AP1900a), rabbit anti-MAVS (1:2000, Enzo Life Science, #ALX-210–929), rabbit anti-IRF3 (1:1000, Abcam, #ab25950), rabbit anti-IRF3-p (1:1000, Cell Signaling, #4947), rabbit anti-PARP (1:1000, Roche, #11835238001), rabbit anti-Caspase7 (1:1000, Cell Signaling, #9492), rabbit-anti CVB3 2C (1:1000, kindly provided by Lindsay Whitten), mouse anti-β actin (1:5000, Sigma-Aldrich, #A5441r,). Respective IRdye680- or IRdye800-conjugated secondary antibodies (LiCOR) were used for detection. For immunofluorescence analysis staining procedures, the mouse anti-dsRNA (1:1000, Jena BioScience, #RNT-SCI-10010), rat anti-Nup98 (1:200, Sigma-Aldrich, #N1038), rabbit anti-hnRNPK (1:250, ThermoFisher, #PA5–19484), goat anti-eIF3η (1:200, SantaCruz, #sc-16377), rabbit anti-G3BP2 (1:200, Bethyl Laboratories, #A1302-040A), goat anti-TIA-1 (1:200, SantaCruz, #sc-1751), rabbit anti-Sam68 (1:200, SantaCruz, #sc-333), antibodies were used. Alexa Fluor 405-, Alexa Fluor 488-, Alexa Fluor 555-, Alexa Fluor 594-, Alexa Fluor 647-conjugated (1:200, Invitrogen) secondary antibodies were used for detection.

**Western blot analysis.** HeLa R19 cells were seeded in 6-well clusters ($5 \times 10^5$ cells/well) and after a 16 hrs recovery cells were infected with recombinant CVB3 at MOI 10 $TCID_{50}$/ml. At the indicated time point, cells were lysed with SDS sample buffer (Tris-HCl pH 6.8, 50mM, Glycerol 1M, Bromophenol Blue 1.5mM, DTT 100mM, SDS 2% w/v). Samples of each lysate were separated by SDS-PAGE in reducing 8% polyacrylamide gels. The proteins were then blotted onto 0.2 µm nitrocellulose membranes by wet electrophoretic transfer. Membranes were washed three times in TBST (20mM Tris, 150mM NaCl + 0.1% Tween-20), 5min each, and incubated in blocking buffer (TBST + 5% BSA) for 1h at RT. Membranes were successively incubated with primary antibodies diluted in blocking buffer for 16 hrs at 4°C, and then for 30min at RT with secondary antibody diluted in blocking buffer. Between and after the incubations, the membranes were washed, thrice each time, with TBST. Finally, membranes were washed once with PBS and scanned using an Odyssey Imager (Li-COR). Densitometric analysis of western blot band intensities was performed using ImageJ. First, a rectangular region of interest was drawn across the entire lane, after which the "plot lanes" functionality was used to generate the intensity profile. Then the area under the curve was calculated for each peak indicating a band. These raw values were corrected for actin band intensity and, if indicated, normalized to mock.

**Immunofluorescence analysis.** HeLa R19 or HeLa-3xNLS-GFP cells were seeded onto 12mm glass coverslips in 24-well plates (Corning Costar) at a density of $5 \times 10^4$ cells per well, grown for 24h and infected with CVB3 recombinant viruses. Virus-containing medium was incubated on the cells for 30min at 37°C and after incubation time replaced with complete DMEM. At the indicated time post-infection, cells were fixed in PBS + 3.7% paraformaldehyde (PFA) for 30min, incubated with PBS + 0.1% glycine for 10min, permeabilized with 0.1% Triton X-100/PBS + S for 10min at room temperature (RT) and blocked in blocking buffer (PBS + 1% BSA + 0.1% Tween-20) for 30min. Cells were then incubated in blocking buffer with primary antibodies for 1h at RT. After washing the cells with PBS + 0.1% Tween-20, cells were incubated with secondary antibody in blocking buffer for 1h at RT. Cells were then washed three times with PBS + 0.1% Tween-20, once with distilled water, and mounted on glass microscope slides in ProLong Diamond Antifade (Invitrogen) mounting medium. Slides were examined by confocal immunofluorescence microscopy (Nikon A1R or Evident SpinSR10 IX83) with a 40x Oil immersion objective (Nikon CFI PlanFLUO NA = 1.3, or Evident UPLAPO NA = 1.4). Sequential laser acquisition was performed (405, 488, 561 and 645 nm) over quadband dichroic filter and bandpass emission filters to separate Alx405/BFP (Nikon: 482/35, Evident: 477/60), for AausFP1/GFP/Alx488 (515/30), Alx555/595 (Nikon: 595/50, Evident: 607/36) and Alx647 (Nikon: 700/75, Evident: 685/40) channels. Images were analyzed by manual scoring in a blinded fashion by two independent observers using ImageJ/FIJI software [76] using the criteria outlined below.

**Quantification of stress granule formation.** Images for SG quantification were obtained as described in immunofluorescence analysis. All Analyses were performed fully blinded to the experimental groups. For each time point and virus condition 15 40x field-of-views (FOVs) were obtained and analyzed as follows. (i) all cells within a FOV were counted based on DAPI stain. cells were excluded if they were partially outside of the FOV. (ii) All dsRNA-positive cells were

marked based on anti-dsRNA J2 antibody staining. (iii) positivity for SGs was assessed based on assessment of G3BP1 and eIF3η signal. SGs were either scored as large or small based on size, and upon the observation that the small early SGs were always negative for eIF3η, whereas the large SGs were always positive. Moreover, cells with small early SGs were mostly negative for dsRNA.

**Live cell imaging and VIRIM.** Hela-R19 AausFP1-STAb (a.k.a. saGFP-STAb) BFP-NLS Clone-1 cells were seeded at $1.5 \times 10^5$ cells per well in a 18-well coverglass-bottom imaging chamber (Ibidi, cat#: 81817) and incubated overnight at 37°C. Cells were infected with indicated 5x-SunTag viruses at MOI of 0.25 in DMEM (10% FCS, 1% P/S) for 30 min at 37°C. The inoculum was removed and replaced with Leibovitz's L15 medium without phenol red (ThermoFisher Scientific, cat#: 21083027) supplemented with 10% FCS and 1% PS. Cells were then imaged on an Evident SpinSR10 IX83 SoRa spinning disc confocal microscope (UPLAPO60xOHR NA = 1.5 or UPLXAPO40xO NA = 1.4 objective, Yokogawa CSU-W1 SoRa, Hamamatsu sCMOS ORCA-Fusion C14440) equipped with a temperature- and $CO_2$-controlled imaging chamber (CellVivo, Evident) using cellSens Dimension software (Evident, vs4.1). Time-lapse images were recorded overnight at 3 min interval with 15 Z-slices of 0.7 μm stepsize, covering the entire cell volume. Several positions were selected randomly throughout the well(s) of interest.

**Image processing and VIRIM analysis.** Before all time-lapse analyses, a maximum intensity projection was performed using Evident CellSens software, and the projections were used for all further analyses. For all intensity analyses, a bleach correction was performed using ImageJ software using the "exponential fit" model. All following image analysis was performed using ImageJ software and recorded/processed in Microsoft Excel and GraphPad PRISM. For all VIRIM analyses, at least 3 independent datasets were obtained and analyzed.

**VIRIM analysis and quantification.** All VIRIM analyses were done as previously described [14,77]. Briefly, VIRIM translation spots were readily distinguishable from background spots and aggregates on basis of mobility, size, and appearance. Each position was screened for analyzable cells based on the following set of inclusion and exclusion criteria. Cells were excluded from analysis if: (i) they had a translation spot in the first timeframe, (ii) they were not entirely within the field-of-view during the entirety of analysis, (iii) they underwent cellular division during analysis, (iv) the first translation spot appeared clearly in a second round of infection (after> 6h), (v) the imaging could not be followed up until phase 5 or a clear phase 2 arrest, (vi) they were double infections (i.e., more than one spot appears in phase 1).

**Definition of VIRIM phases.** Phases were called upon observation of at least 4 consecutive frames in which a spot was present or absent. Phase 1, translation of the incoming vRNA, was defined as the initial appearance of a single VIRIM translation spot. After a while, this initial spot disappears, marking the start of phase 2, the first round of replication. Next, one of three things can happen; (i) there is a reappearance of multiple new translation spots, marking successful replication and the start of phase 3, (ii) there is no reappearance of any spots, signifying replication failure and infection arrest, or (iii) there is reappearance of a single spot, signaling a replication failure, and recommencement of translation on the initial incoming vRNA. The latter scenario can occur in a recurring pattern of translation followed by replication, which are called translation pulses and breaks, which finally results in either successful start of phase 3 or phase 2 arrest. Here, all translation breaks and pulses up until the final replication break before success or failure were collectively called phase 1. Phase 3 constitutes the initiation of translation on nascent vRNAs synthesized during initial replication, and is marked by a steep increase in the number of translation spots. Once this stabilizes at a constant number, phase 4 is called, which is manifested by continued translation and a new round of replication. The end of phase 4 is marked by a second steep increase in the number of VIRIM translation sites, and this is called phase 5. Once the number of spots exceeds 50, we are no longer able to confidently distinguish between individual spots, marking the end of our analysis.

**Quantification of VIRIM translation site intensity.** To identify the intensity of VIRIM translation sites we measured the mean intensity in a square region of interest (ROI) of 8x8 pixels centered on the translation spot. To correct for background and STAb expression levels, we subtracted the average intensity of three randomly selected 8x8 pixel ROIs on surrounding cytosol of the same cell. If spots appeared very close together, or were spherical (indicating this is likely the same spot moving between

multiple Z-frames), they were excluded from analysis. For phase 1 spot intensity we measured the spot intensity up to the first 10 frames (with at least 5, else it was excluded from analysis), and averaged this. For phase 4, the first two or three frames were measured, with maximum 5 spots per frame, for a total of 10 spots. If more than 5 spots were present in one frame, spots were selected starting from the upper-left side of the cell, and working to the right and down, until 5 frames were measured. The lower VIRIM spot intensity we observed for CVB3-wt compared to CVB3–3 CDcs-2Awt can be explained by the replacement of the 2A$^{pro}$ with the 3 CD$^{pro}$ cleavage site at the P1-P2 junction, as this means that the nascent polypeptide chain, and consequently the associated STAb-GFP, will stay attached to the translating ribosome for a longer time (Figs 8B and S6B).

**Quantification of nuclear leakage of NLS-BFP.** Only those infections where at least 10 frames prior to the start of phase 1 could be analyzed were selected. Measurements were done starting 10 frames prior to the start of phase 1, or in uninfected cells, at the start of the time-lapse movie. For each frame, three 5x5 ROIs were randomly selected in the cytosol, and 3 5x5 ROIs were randomly selected in the nucleus (avoiding the nucleoli). Both the edges of the cell, as well as the edges of the nucleus were excluded. The cytosol also contained areas where signal was consistently less than in surrounding cytosol, likely representing organelles. These were also excluded. The BFP intensity was measured for each frame, and the three ROIs were averaged. The mean cytoplasmic BFP intensity was substracted in each frame from the nuclear BFP intensity and this was normalized to the average of the 3 frames preceeding appearance of the first spot OR the first 3 frames in a movie if uninfected.

***In vivo* metabolic labelling.** Newly synthesized proteins were determined by metabolic labeling using a threonine-derived non-canonical amino acid β-ethynylserine (βES), followed by fluorescence labeling of βES based on an adapted protocol of the published THRONCAT assay [36]. In particular, HeLa R19 cells were seeded in 6-well plates ($4.0x10^5$ cells/well) and after a 16 hrs recovery they were infected with recombinant CVB3 at MOI 100 TCID$_{50}$/ml. After 1h, viral inoculum was replaced with complete DMEM. One hour prior to cell lysis, cells were incubated with 2μM (βES) in complete DMEM at 37°C for 1h and either left untreated or treated with 500 μM sodium arsenite (NaAsO2, Riedel-de Haen) to generate a positive control for translation arrest. Following incubation, at the indicated time point, cells were washed with PBS, lysed with lysis buffer (HEPES 50 mM pH 7.4 (Sigma-Aldrich), NaCl 150 mM (ThermoFisher), Triton-X-100 1% (Sigma-Aldrich), 1×EDTA-free protease inhibitor cocktail (Roche)), and subjected to sonication on ice. Cell lysates were cleared by centrifugation at 4°C, after which the supernatant was transferred to a new tube and the protein concentration was determined by BCA assay (Micro BCA Protein Assay kit, ThermoFisher). Lysates were flash-frozen and stored at -80°C until further usage. Cy5 conjugation to HeLa proteome via azide-alkyne click chemistry was performed as described by Ignacio et al., 2023 [36]. Protein samples were run on an SDS-PAGE and Image Quant800 Imager (Cytiva Amersham) was used for image acquisition.

**Statistical analysis.** Data display and analysis were performed using GraphPad Prism version 10 (GraphPad Software, CA). Appropriate statistical analyses were performed as indicated using GraphPad Prism 10. Unless otherwise indicated, all statistical tests assumed normal distribution of data and were done using an α-value of 0.05 as cutoff for statistical significance.

**Software.** Figures were made in Adobe Illustrator 2023. Most of the schematics were produced using BioRender.com.

## Supporting information

**S1 Data. Densitometric analysis of WB band intensities.** Band intensities were quantified using ImageJ as detailed in the Methods section. Band intensities were corrected for Actin (see actin normalization sheet). If indicated, the values were normalized to mock conditions.
(XLSX)

**S1 Fig. Growth kinetics of 2Awt and 2Amut viruses on different cell lines.** (A) Assessment of viral polyprotein processing efficiency by western blot analysis of CVB3 VP1 protein and β-actin. HeLa-R19 cells, (mock-)infected with CVB3, 3 CD$_{cs}$-2Awt, T2A-2Awt at an MOI of 10, were lysed at 2, 4, 6, 8 and 10 hpi. (B) CVB3, 3 CD$_{cs}$-2Awt, and 3 CD$_{cs}$-2Amut

viral stocks were titrated on HeLa-R19, BGM, Vero E6 and A549 by end-point dilution. (C) Growth curves of CVB3, T2A-2Awt and T2A-2Amut viruses in HeLa-R19, BGM, Vero E6 and A549 cells performed as in Fig 1E. Data represent the mean ± SEM of the three technical replicates (B and C). Related to Fig 1.
(TIF)

**S2 Fig. 2A^pro expression levels in 3 CD_cs-2Amut infected cells.** Western blot analysis for CVB3 2A^pro. HeLa-R19 cells, (mock-)infected with CVB3, 3 CD_cs-2Awt, or 3 CDc_cs-2Amut at an MOI of 10, were lysed at 2, 4, 6, 8 and 10 hpi. Related to Fig 3A.
(TIF)

**S3 Fig. Cleavage of PABP at later time points and SGs in infected cells stained with TIA-1 and Sam68.** (A) Western blot analysis for PABP. Cell lysates derive from the same experiment shown in Fig 4D. (B) IF analysis of HeLa-R19 cells infected with CVB3, 3 CD_cs-2Awt and 3 CD_cs-2Amut viruses, fixed and stained for dsRNA as infection marker and TIA-1 and Sam68 as SG marker. Experiment was performed as described in Fig 4C. Magnified regions are shown in white boxes. Scale bar: 25 μm. Related to Fig 4.
(TIF)

**S4 Fig. Western blot analysis of markers of apoptosis.** (A) Western blot analysis of caspase 7 (Casp7) and PARP using cell lysates corresponsing to the same experiment shown in Fig 4D. Black triangles mark intact proteins, white triangles the corresponding cleavage products. Related to Fig 5.
(TIF)

**S5 Fig. MAVS cleavage products generated by 3C^pro and 2A^pro in CVB3–2Awt and CVB3–2Amut infected cells.** (A) Western-blot analysis of MAVS in cell lysates obtained from the same experiment shown in Fig 4D. Black triangles denote intact proteins, white triangles the cleavage products. (B) non-normalized dataset as represented in Fig 6C. Statistical significance was assessed by multiple two-tailed unpaired t-tests with multiple comparisons corrections using the Bonferroni-Dunn method (* $p < 0.05$; ** $p < 0.01$; *** $p < 0.001$). Related to Fig 6.
(TIF)

**S6 Fig. 2A^pro rapidly and exclusively triggers nucleocytoplasmic trafficking disorder.** Hela-R19 saGFP-STAb BFP-NLS C1 cells were infected at an MOI of 0.25 with ST-CVB3–2Awt or ST-CVB3–2Amut and time-lapse imaging of STAb-GFP and BFP-NLS was performed. (A) Representative pictures of cells infected with ST-CVB3–2Awt or -2Amut at different time points aligned to the start of phase 1. (B) average normalized BFP-NLS intensity ratio between nucleus and cytoplasm over time aligned to start phase 1. Scale bar: 10 μm. (C) Kaplan-Meier curves depicting duration of phase 1 in those cells with successful replication, non-successful replication (i.e., phase 2 arrested), or both (total). (D) Fraction of infections with indicated number of translation pulses. Related to Fig 7.
(TIF)

**S7 Fig. VIRIM spot intensity as a measure of viral translation efficiency.** (A) Schematic representation of VIRIM spot intensity analysis. Upper panel shows a situation with comparatively lower translation efficiency, less translating ribosomes on a single vRNA, and consequently a smaller accumulation of fluorescence signal. The lower panel depicts the scenario of a comparatively higher translation efficiency. (B) Absolute intensity measurements of different CVB3 mutants. The higher intensity for 3 CDcs-2Awt is the consequence of a relatively longer effective transcript length (i.e., the length of transcript that is decoded before the nascent peptide chain is released from the ribosome). statistical significance was assessed by a one-way analysis of variance (ANOVA) with multiple comparisons testing. * $p < 0.5$; ** $p < 0.01$; *** $p < 0.001$. (A) created in BioRender. Schipper, J. (2025) https://BioRender.com/unbzo59. Related to Fig 8.
(TIF)

**S8 Fig. Overview of the diverse functions of 2A^pro and 3C^pro.** Schematic depiction of the different functions of 2A^pro and 3C^pro. (1) 2A^pro is critical for establishing the host translation shutdown observed in enterovirus-infected cells, via cleavage of eIF4G and PABP. Even though 3C^pro was previously also found to cleave PABP, we were not able to detect this cleavage during CVB3–2Amut infection. (2) 2A^pro, and not 3C^pro, is essential for the rapid establishment of a nucleocytoplasmic trafficking disorder (NCTD), via the cleavage of nucleoporins. We can however not fully exclude that 3C^pro might also subtly contribute to the induction of the NCTD. (3) 2A^pro catalytic activity is required for preventing the formation of stress granules (SGs) during enterovirus infection, although the mechanism remains unclear. G3BP1 cleavage by 3C^pro does not seem to be required for SG inhibition. (4) 2A^pro cleaves MAVS and MDA5. Interestingly, we were able to detect 3C^pro-mediated cleavage of MAVS, which was previously reported, only at much later timepoints. Moreover, the temporal dynamics of MAVS and MDA5 cleavage raise questions about their relevance in preventing the induction of IFN signaling. We hypothesize that the 2A^pro-mediated NCTD might play a previously under-appreciated role in suppressing IFN signaling. Even though from our study it appears that 3C^pro is less important than 2A^pro for the depicted effects, we cannot exclude that 3C^pro might still contribute. Created in BioRender. Schipper, J. (2025) https://BioRender.com/fexc61o. (TIF)

## Author contributions

**Conceptualization:** Jelle G. Schipper, Chiara Aloise, Frank J.M. van Kuppeveld.

**Data curation:** Jelle G. Schipper, Chiara Aloise, Sereina O. Sutter, Marvin E. Tanenbaum, Frank J.M. van Kuppeveld.

**Formal analysis:** Jelle G. Schipper, Chiara Aloise, Sereina O. Sutter, Marleen Zwaagstra, Rana Abdelnabi, Dagmar Roelofs, Judith M.A. van den Brand.

**Funding acquisition:** Sereina O. Sutter, Johan Neyts, Marvin E. Tanenbaum, Frank J.M. van Kuppeveld.

**Investigation:** Jelle G. Schipper, Chiara Aloise, Sereina O. Sutter, Marleen Zwaagstra, Arno L.W. van Vliet, Rana Abdelnabi, Dagmar Roelofs, Judith M.A. van den Brand.

**Methodology:** Jelle G. Schipper, Chiara Aloise, Sereina O. Sutter, Rana Abdelnabi, Judith M.A. van den Brand, Richard W. Wubbolts, Lucas J.M. Bruurs, Hendrik Jan Thibaut, Johan Neyts, Marvin E. Tanenbaum, Frank J.M. van Kuppeveld.

**Project administration:** Marvin E. Tanenbaum, Frank J.M. van Kuppeveld.

**Resources:** Bob Ignacio, Kimberly M. Bonger, Richard W. Wubbolts, Lucas J.M. Bruurs.

**Supervision:** Hendrik Jan Thibaut, Johan Neyts, Marvin E. Tanenbaum, Frank J.M. van Kuppeveld.

**Validation:** Jelle G. Schipper, Chiara Aloise.

**Visualization:** Jelle G. Schipper, Chiara Aloise.

**Writing – original draft:** Jelle G. Schipper, Chiara Aloise, Sereina O. Sutter, Frank J.M. van Kuppeveld.

**Writing – review & editing:** Jelle G. Schipper, Chiara Aloise, Sereina O. Sutter, Marleen Zwaagstra, Arno L.W. van Vliet, Rana Abdelnabi, Bob Ignacio, Kimberly M. Bonger, Dagmar Roelofs, Judith M.A. van den Brand, Richard W. Wubbolts, Lucas J.M. Bruurs, Hendrik Jan Thibaut, Johan Neyts, Marvin E. Tanenbaum, Frank J.M. van Kuppeveld.

## Acknowledgments

We thank all members of the Virology section at the Faculty of Veterinary Medicine of Utrecht University for helpful discussions.

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
