## [Decision Letter · Decision Letter 0]

26 May 2025

PPATHOGENS-D-25-00714

The multifaceted role of the viral 2A protease in enterovirus replication and antagonism of host antiviral responses.

PLOS Pathogens

Dear Dr. van Kuppeveld,

Thank you for submitting your manuscript to PLOS Pathogens. After careful consideration, we feel that it has merit but does not fully meet PLOS Pathogens's publication criteria as it currently stands. Therefore, we invite you to submit a revised version of the manuscript that addresses the points raised during the review process.

Please submit your revised manuscript within 30 days Jul 25 2025 11:59PM. If you will need more time than this to complete your revisions, please reply to this message or contact the journal office at plospathogens@plos.org. Please include the following items when submitting your revised manuscript:

We look forward to receiving your revised manuscript.

Kind regards,

Eric Jan, Ph.D.

Guest Editor

PLOS Pathogens

Michael Letko

Section Editor

PLOS Pathogens

Sumita Bhaduri-McIntosh

Editor-in-Chief

PLOS Pathogens

orcid.org/0000-0003-2946-9497

Michael Malim

Editor-in-Chief

PLOS Pathogens

orcid.org/0000-0002-7699-2064

**Additional Editor Comments:**

Your manuscript was reviewed by three experts in the field. All three reviewers all agree that this study is of merit with important advances. However, they have provided comments that will improve the manuscript. Please address all of the comments in a revised manuscript.

**Journal Requirements:**

At this stage, the following Authors/Authors require contributions: Jelle G. Schipper, Chiara Aloise, Sereina O. Sutter, Marleen Zwaagstra, Arno L.W. van Vliet, Rana Abdelnabi, Bob Ignacio, Kimberly M. Bonger, Dagmar Roelofs, Judith M.A. van den Brand, Richard Wubbolts, Lucas J.M. Bruurs, Hendrik Jan Thibaut, Johan Neyts, Marvin E. Tanenbaum, and Frank J.M. van Kuppeveld. Please ensure that the full contributions of each author are acknowledged in the "Add/Edit/Remove Authors" section of our submission form.

Potential Copyright Issues:

- Figure 2. Please confirm whether you drew the images / clip-art within the figure panels by hand. If you did not draw the images, please provide (a) a link to the source of the images or icons and their license / terms of use; or (b) written permission from the copyright holder to publish the images or icons under our CC BY 4.0 license. Alternatively, you may replace the images with open source alternatives. See these open source resources you may use to replace images / clip-art:

5) Please amend your detailed Financial Disclosure statement. This is published with the article. It must therefore be completed in full sentences and contain the exact wording you wish to be published. Please ensure that the funders and grant numbers match between the Financial Disclosure field and the Funding Information tab in your submission form. Note that the funders must be provided in the same order in both places as well. State what role the funders took in the study. If the funders had no role in your study, please state: "The funders had no role in study design, data collection and analysis, decision to publish, or preparation of the manuscript.".

**Reviewers' Comments:**

Reviewer's Responses to Questions

**Part I - Summary**

Reviewer #1: This study investigates the role of the enteroviral 2A protease in viral replication and immune evasion, using engineered coxsackievirus B3 (CVB3) with inactive 2A via alternative cleavage sequence on viral genome. The authors demonstrate that 2A is critical for efficient replication in vitro and in vivo, and for disrupting host defenses including nucleocytoplasmic transport, mRNA translation, stress granule formation, and interferon responses. Using single-molecule live-cell imaging, they also show that 2A is required for the early replication of incoming viral RNA.

The study cleverly addresses a major experimental challenge by enabling the functional analysis of the 2A protease in the context of infection. The approach is innovative and yields valuable insights into the multifaceted role of 2A in viral replication and host interaction. However, several points require clarification or additional data to strengthen the conclusions:

Reviewer #2: In this manuscript van Kuppeveld and colleagues have queried the essential roles of enterovirus 2A proteinase in the initial establishment of infection. Analogous to prior work with mengovirus, the authors created novel variant CVB3 viruses to test downstream roles of 2Apro after the most essential role of 2Apro, to cleave the capsid domain away from nonstructural proteins co-translationally, was mimiced by three approaches. These were inserting an peptide sequence with an 3Cpro cleavage site, inserting a self cleaving peptide from and insect virus, or a second IRES from Theilers virus, all to to separate capsid from nonstructural proteins. In addition, three point mutations were inserted to catalytically inactivate 2Aproteinase. The mutant viruses without 2Apro were then compared in a series of experiments versus identical virus with active 2Apro or WT virus to assess the role of the protease in the virus life cycle.

The three constructs produced a virus that replicates but was modestly defective in single step growth curves in HeLa cells, achieving slightly lower virus replication with similar kinetics. Infections on other cell lines revealed greater apparent replication defects, the three variants were similar to each other. The authors then completed all the remaining work with just one CVB3 variant containing the extra 3Cpro cleavage site.

Through several figures the author test and find large replication and pathogenesis defects in virulence in an SJL mouse model for pancreatitis. The authors then turn back to HeLa cells for remaining analysis.

The next series of mutant 2Apro/replication defects described can be ascribed to functions previously associated with 2Apro in many publications. Many of the findings have been reported previously using other approaches, including blockage of nucleocytoplasmic transport, translation shutoff, hampering stress granule formation, apoptosis and the IFN response. The authors include an interesting experiment showing that IFN paracrine signaling can be overcome by 2Apro. Much of this work is confirmatory, but not new, however confirmations by a new approach is valuable.

The authors use a single-molecule live imaging system to demonstrate the importance of 2Apro in the initial round of replication. In many ways the most interesting part of the manuscript is in Figs 7 and 8 where the single cell analysis system VIRIM is utilized. This is a two-component system consisting of cells expressing a single-chain variable antibody fragment fused to a fluorescent protein (SunTag-antibody, STAb-FP) and viruses that have been genetically engineered to express a short suntag peptide array at the N-terminus of the viral polyprotein. The authors used this approach to separate the initial translation phase from replication and secondary translation/replication phases of infection in single cells and this approach can be powerful.

In these experiments the authors show that 2Apro facilitates the primary round of virus replication, though no direct evidence for rescue of initial (phase 1) virus translation by 2Apro was shown. The authors showed by following intensity of VIRIM translation suntag spots, that 2Apro progressively stimulates viral translation efficiency as the infection proceeds.

The authors do point out the limitations of analysis of 2Apro vs 3Cpro functions in the discussion in regards to regulation of stress granule (formation and resolution), where roles for both proteins have been reported.

Overall the manuscript is well written, methodically laid out and complete. The approach using 2Apro-deficient viruses show that 2Apro is critical for relieving replication bottlenecks in the initial round of replication of infecting virus RNA, subsequently blocking antiviral responses and promoting favorable intracellular conditions for replication. Multiple mechanistic aspects of the replication cycle and virus-host interactions were interogated. The use of a novel virus construct approach largely corroborates many prior findings in the field, but adds some new twists. The single cell replication approach here is novel and could be powerful.

Reviewer #3: This manuscript describes in detail the numerous aspects of enterovirus 2A protease in the infectious cycle. A variety of experimental and novel technological approaches are being used to study the function of 2A in viral genomes that carried an IRES instead of active 2A or T2A cleavage sites. It was noted that 2A is important for virus virulence of coxasckievirus B3 in a mouse model, is essential for disrupting nucelocytoplasmic transport, stress granule formation and overcoming interferon-induced restriction factors. Finally, it was discovered that 2A is essential during the initial rounds of replication.

**Part II – Major Issues: Key Experiments Required for Acceptance**

Reviewer #1: 1. Statistical analyses are missing in several figures, such as Figures 1, 2C and 6C. Please provide appropriate statistical evaluations to support the conclusions.

2. Quantification of Western blot data is lacking. Please include densitometric analysis for all Western blots presented in Figures 3A, 4A–B, 4D, and 6B to allow for proper assessment of protein expression and cleavage efficiency. For instance, in lines 347-349, the statement “Interestingly, we observed earlier and more pronounced PKR and eIF2α phosphorylation in CVB3-2Amut- than in CVB3-2Awt-infected cells (Figure 4A) …” is not fully supported by the results in the absence of data quantification.

3. In Figure 1B, can the authors detect the P1 and P2 cleavage fragments directly by Western blot to assess the efficiency of polyprotein processing among the three recombinant CVB3 viruses containing WT-2A? The observation that viral RNA levels remain unchanged while viral titers are greatly reduced between WT-CVB3 and mutant-CVB3 (Figure 1B), in contrast to the opposite trend seen in Figure 2B, needs further clarification beyond the possibility of inefficient polyprotein cleavage. Moreover, could the authors comment on the typical duration of a single replication cycle for these viruses in HeLa cells? Is it completed within 16 hours?

4. In Figure 1E, the authors should discuss why the 3CD-2Amut virus produced much lower viral titers in BGM, Vero E6, and A549 cells compared to 3CDcs-2Awt, while this difference was less pronounced in HeLa R19 cells. What factors might account for the cell-type-dependent differences in viral replication or susceptibility?

5. Figure 4A, data for PKR cleavage is not convincing. The authors should show cleavage of PKR total protein which looks unchanged across the different infections.

6. In Figure 4B, viral protein translation is clearly observed in the 3CDcs-2Awt group at 5–6 hours post-infection. However, by 7 hours and beyond, the viral protein bands appear to diminish or disappear. Could the authors clarify the reason for this decline? This point is also relevant to Question 3 regarding the duration of the viral life cycle in HeLa cells.

7. Figure 6C. There is a difference at 0 units of IFN-�2 to begin with, is the data normalized to this difference/how significant is the data for increased concentration?

Reviewer #2: None noted

Reviewer #3: (No Response)

**Part III – Minor Issues: Editorial and Data Presentation Modifications**

Reviewer #1: 1. Line 39, revise to “The roles of viral proteins in these processes have been studied……”

2. Line 243, define “NLS, M9-NLS, RS-NLS, and NES”.

3. Line 267, correct the typo of “CBV3 and CBV3-2Awt” to “CVB3 and CVB3-2Awt”

4. In Figure 6B, the p-IRF3 blots appear to be significantly cropped. Could the authors provide the full, un-cropped images for transparency and clarity?

5. Figure 5. Why was 16hpi omitted for 3CDcs2Awt?

6. Line 390, correct the typo of “indictaed”.

7. Line 451 and 252. Omit “both”.

8. Line 508-509, Figure 7E does not show phase 5.

Reviewer #2: Conclusions of roles of 2A vs 3Cpro in SG formation is a bit strong. The 2Apro defective mutants have seriously delayed kinetics of 3Cpro cleavage of G3BP compared to WT virus. This could be a clue to a role for other 2Apro functions providing a modified G3BP1 substrate for 3Cpro cleavage, ie after ribosome runoff, loss of interactions with eIF4G or other proteins, etc. Interestingly, a similar delay in kinetics of 3Cpro cleavage of MAVS was also seen with the 2Amut virus and could be discussed.

Reviewer #3: Comments:

1. Show the abundances of 2A by Western blot analyses.

2. Are cleavges of cellular transciption factors part of 2A’s function?

3. Discuss potential dominant-negative effects of 2A on the observed phentypes.

4. Adding a Table that lists the functions of proteinases 3C and 2A would be helpful.

PLOS authors have the option to publish the peer review history of their article (what does this mean? ). If published, this will include your full peer review and any attached files.

**Do you want your identity to be public for this peer review?** For information about this choice, including consent withdrawal, please see our Privacy Policy .

Reviewer #1: **Yes: ** Honglin Luo

Reviewer #2: No

Reviewer #3: No

**Figure resubmission:**
---

## [Decision Letter · Decision Letter 1]

11 Aug 2025

Dear Prof. Dr. van Kuppeveld,

We are pleased to inform you that your manuscript 'The multifaceted role of the viral 2A protease in enterovirus replication and antagonism of host antiviral responses.' has been provisionally accepted for publication in PLOS Pathogens.

Best regards,

Eric Jan, Ph.D.

Guest Editor

PLOS Pathogens

Michael Letko

Section Editor

PLOS Pathogens

Sumita Bhaduri-McIntosh

Editor-in-Chief

PLOS Pathogens

orcid.org/0000-0003-2946-9497

Michael Malim

Editor-in-Chief

PLOS Pathogens

orcid.org/0000-0002-7699-2064

Thank you for your thorough responses to reviewers' comments and the additional experiments that further supports the conclusions of the manuscript.

Reviewer Comments (if any, and for reference):

Reviewer's Responses to Questions

**Part I - Summary**

Reviewer #1: The authors have satisfactorily addressed my previous concerns in this revision.

**Part II – Major Issues: Key Experiments Required for Acceptance**

Reviewer #1: (No Response)

**Part III – Minor Issues: Editorial and Data Presentation Modifications**

Reviewer #1: (No Response)

PLOS authors have the option to publish the peer review history of their article (what does this mean? ). If published, this will include your full peer review and any attached files.

**Do you want your identity to be public for this peer review?** For information about this choice, including consent withdrawal, please see our Privacy Policy .

Reviewer #1: No

---

## [Editor Report · Acceptance letter]

Dear Prof. Dr. van Kuppeveld,

We are delighted to inform you that your manuscript, " 

The multifaceted role of the viral 2A protease in enterovirus replication and antagonism of host antiviral responses.," has been formally accepted for publication in PLOS Pathogens.

Best regards,

Sumita Bhaduri-McIntosh

Editor-in-Chief

PLOS Pathogens

orcid.org/0000-0003-2946-9497

Michael Malim

Editor-in-Chief

PLOS Pathogens

orcid.org/0000-0002-7699-2064